

# Elasto-plastic-adhesive DEM model for simulating hillslope debris flows: cross comparison with field experiments

Adel Albaba[1], Massimiliano Schwarz[1], Corinna Wendeler[2], Bernard Loup[3], and Luuk Dorren[1]

[1]Bern University of Applied Sciences, School of Agricultural, Forest and Food Science HAFL, Länggasse 85, 3052 Zollikofen, Switzerland
[2]Geobrugg Protection Systems, Aachstrasse 11, 8590 Romanshorn, Switzerland
[3]Federal Office for the Environment (FOEN), Papiermühlestrasse 172, 3063 Ittigen, Switzerland

**Correspondence:** Adel Albaba (adel.albaba@bfh.ch)

**Abstract.** This paper presents a Discrete Element-based elasto-plastic-adhesive model which is adapted and tested for producing hillslope debris flows. The numerical model produces three phases of particle contacts: elastic, plastic and adhesion. The model capabilities of simulating different types of cohesive granular flows were tested with different ranges of flow velocities and heights. The basic model parameters, being the basal friction ($\phi_b$) and normal restitution coefficient ($\epsilon_n$), were calibrated

using field experiments of hillslope debris flows impacting two sensors. Simulations of 50 $m^3$ of material were carried out on a channelized surface that is 41 m long and 8 m wide. The calibration process was based on measurements of flow height, flow velocity and the pressure applied to a sensor. Results of the numerical model matched well those of the field data in terms of pressure and flow velocity while less agreement was observed for flow height. Those discrepancies in results were due in part to the deposition of material in the field test which are not reproducible in the model. A parametric study was conducted to

further investigate that effect of model parameters and inclination angle on flow height, velocity and pressure. Results of best-fit model parameters against selected experimental tests suggested that a link might exist between the model parameters $\phi_b$ and $\epsilon_n$ and the initial conditions of the tested granular material (bulk density and water and fine contents). The good performance of the model against the full-scale field experiments encourages further investigation by conducting lab-scale experiments with detailed variation of water and fine content to better understand their link to the model's parameters.

## 1  Introduction

Worldwide, the growing demand for land to build led to the urbanization of mountainous areas. This increased the importance of studying the different processes of natural hazards that impose danger to residential area and infrastructure. On steeper slopes in the affected areas, gravity driven mass movements such as shallow landslides, triggered by intense rainfall or earthquakes, are frequent. In Switzerland, shallow landslides and hillslope debris flows (Fig.1) are yearly responsible for high infrastructure

damage, blocking of important highways, evacuations and deaths (Andres and Badoux, 2018). Moreover, these processes could increase the damage caused by floods by clogging channels and rivers at bridges and passages. Hillslope debris flows are one type of mass movements where shallow landslides transform into an unconfined (unchannelized) flows following heavy rainfalls or earthquakes. They are sometimes referred to as debris avalanche, but unlike the ones described by Hungr




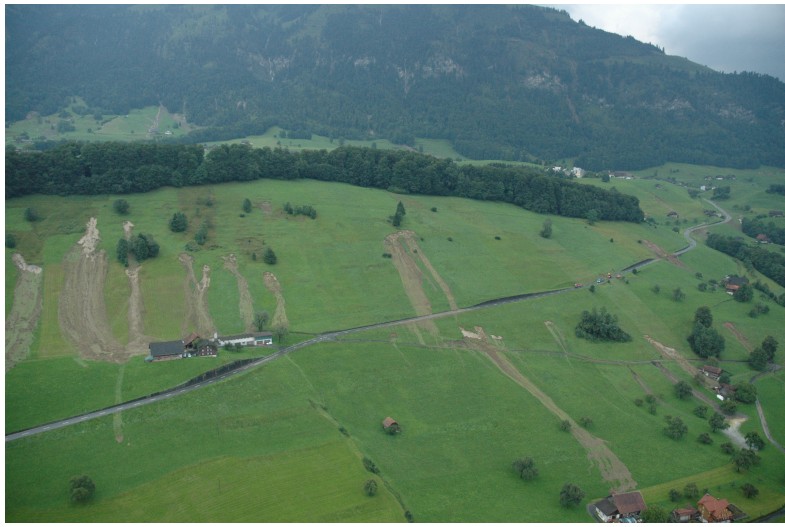

**Figure 1.** Aerial photo of several hillslope debris flow events in Switzerland following severe rainfall event in summer 2005 (source: Federal office for the environment FOEN, 2005)

et al. (2014), they rarely entrain sediments along their way (Hürlimann et al., 2015). In comparison with channelized debris flows, they tend to have shorter run-out distance due to their lateral spreading as no confinement exists. Their overall assessment comprises: (i) the mechanics of initial slope failure (Jibson, 1995; Schwarz et al., 2010; Olivares and Picarelli, 2003; Klubertanz et al., 2009; Shen et al., 2017). (ii) the transformation from a sliding block into a deformed flowing mass (Iverson et al., 1997; Gabet and Mudd, 2006). (iii) the kinematics of the flowing mass (velocity, run-out, etc.). While both first aspects have been extensively investigated, the kinematics of hillslope debris flows have been rarely investigated.

Numerical modeling has been deployed as an effective tool in simulating the behavior of shallow landslides and hillslope debris flows (Hungr, 1995; Montrasio and Valentino, 2016; Ran et al., 2018). For example, the software RAMMS:Hillslope was developed in the WSL, based on a momentum balance using a Voellmy rheological model (Voellmy, 1955; Christen et al., 2010; Graf and McArdell, 2011). Another example is the mass balance-based Flow-R model which was developed in the University of Lausanne primarily for regional susceptibility assessments of debris flows (Horton et al., 2013). The model was successfully applied to different case studies in various countries with variable data quality. It was also found relevant to assess other natural hazards such as rockfall, snow avalanches and floods. The basic concept of Flow-R was recently implemented and extended in a model developed at the HAFL called M-Flow, which has been tested for modeling hillslope debris flows (Scherer, 2016). M-Flow model is fairly simple and only accounts for the mass distribution of the flowing mass according to the terrain, without in-depth investigation of the physical aspects of hillslope debris flows. However, first preliminary tests using M-Flow to reproduce the propagation and impact pressures of real hillslope debris flow events gave promising results.

Discrete element simulations were also used to investigate flowing characteristics and impact pressures of granular flows down inclines (Teufelsbauer et al., 2009; Albaba et al., 2015; Wu et al., 2016; Shen et al., 2018a). Parameters such as flowing





velocity, flowing height and impact pressure were characterized in these simulations at different sections with in-detail investigation. Results of these simulations were often compared to depth-averaged hydrodynamic models concerning the impacting pressure of these flows on rigid barriers (Faug, 2015; Albaba et al., 2018). Moreover, parametric studies investigating the effect of inclination angles of the chute and the barrier on flow behavior and impact pressure were also investigated (Albaba,

2015). Although these simulations agreed well with the proposed theory concerning the impact, they were mainly carried out for dry granular flows with no consideration of fluid presence. Such presence would change the flow characteristics and the time-history of its applied pressure (Vollmöller, 2004; Kattel et al., 2018). Other DEM-based models accounted for the presence of fluid by coupling DEM with either LBM (Lattice-Boltman-Method) or a CFD (Computational Fluid Dynamics) solver (Leonardi et al., 2016; Ding and Xu, 2018). Such models were promising for theoretical research questions but are

computationally expensive for practical use in the daily practice of hillslope debris flow hazard assessment.

In addition to numerical modeling, lab, medium and full-scale experiments were carried out to investigate the flowing mass behavior of debris flows and their impact on rigid objects. For instance, Hürlimann et al. (2015) set up a 7.5 meter long experimental chute in which different samples with different grain size distributions, water contents and volumes were tested. It was found that increasing water content, even by a small amount, would greatly increase the run-out distance (exponential

relationship). The increase of clay content resulted in a decrease in the run-out distance. In addition, a proportional relationship was observed between the run-out distance and the volume, although the effect was rather small.

An intermediate-scale flume for testing debris flow was built by the United States Geological Survey (USGS) which is 95 m long, 2 m wide and 1.2 m deep Iverson and LaHusen (1993). The majority of its length slopes $31°$ while the remaining part (7 m) gradually flattens to $2.5°$ and it has a loading capacity of 20 $m^3$ of granular material. The flume has been actively used

since 1993 for investigating different aspects of debris flows physics.

A full-scale hill-slope debris flow experiment was carried out by Bugnion et al. (2012) by measuring the impact of 16 events of 50 $m^3$ in volume on two impact sensors. A 41-m-long, 8-m-wide channel was constructed on the side of a rock quarry in Switzerland. The advantage of such full-scale investigation is that it allowed for detailed measurements that are usually overlooked in the post-analysis of previous landslides and hillslope debris flows, especially those parameters that are very

difficult to estimate for the post-analysis of events, mainly flow height, velocity and and impact pressure.

All in all, although advancements have been achieved, the understanding of shallow landslides and hillslope debris flows is still lacking. Assessments of such natural hazards, unlike rockfalls and avalanches, is therefore mostly based on experience of experts. In order to improve the hazard assessment of shallow landslides, new tools and methods are needed to calculate the disposition, the evolution as well as the run-out of hillslope debris flows on a slope for different situations (normal situation,

severe precipitation, with and without forest cover, etc.). This paper presents a new DEM model analysis, based on the work of Luding (2008), to model the flow dynamics of hillslope debris flows down inclined planes. First, the DEM model is described in detail highlighting the key parameters to calibrate. Subsequently, the field experimental data that are used to calibrate and validate the model are presented. Next, a cross comparison between model results and experimental data is carried out, with a special focus on flow height, flow velocity and the pressure applied by the flow on pressure measuring sensors. A parametric


study is then presented investigating the effects of varying the model microscopic parameters in addition to the mean particle diameter ($d_{50}$) and channel inclination angle ($\alpha$). Finally, the main results are discussed and conclusions are drawn.

## 2 Materials and Methods

### 2.1 Field experiment of hillslope debris flow

A series of full-scale field experiments of hillslope debris flow were carried out between 2008 and 2010 at the Veltheim test site in the Canton Aargau in Switzerland (cf. Bugnion et al. 2012). The objectives of the experiments were to measure the height and velocity of hillslope debris flows, as well as the pressure. A flexible barrier, which is widely used as a protection measure against many types of mass movements (Volkwein, 2005; Brighenti et al., 2013; Albaba et al., 2017), was installed downslope the channel and was intensively instrumented in order to measure the internal forces and deformations developed in the barrier

while being impacted by flows. In total, 16 tests were carried out by which 8 tests consisted of one release while the remaining 8 tests consisted of successive releases, where the debris flows were released successively without cleaning the channel. Each release had a constant volume of 50 $m^3$.

### 2.1.1 Geometry and material used

At Veltheim, a 41-m-long, 8-m-wide channel was constructed on the side of a rock quarry in Switzerland (Fig.2). The channel

was excavated to the bedrock bottom creating side walls of 1 meter. This excavated material was used to prepare the granular flow which led to slight differences in granular size distribution of tested materials. In addition, different levels of water content were added to the tested material ranging from 14-28% which created densities between $1760 - 2110 \ kg/m^3$. The channel was instrumented with laser distance sensors at distances of 14 and 26 m from the reservoir gate. In addition, two pressure plates (2-kHz signal capacity) were installed at a distance of 30 m from the reservoir gate with dimensions of 160 mm x 225 mm and

240 mm x 295 mm for the small and large plate respectively. The tested flow was similar to debris flows generated in weakly channelized channels. Figure 3 shows snapshots of test 10 shortly after release (3 seconds after release) and after impacting the pressure sensors (6 seconds after release).

### 2.1.2 Measured parameters in the experiments

For each released flow, the flow height at sections 1 and 2 were measured using laser sensors. Sections 1 and 2 are located

at distances 14 m and 26 m respectively downstream from the starting reservoir, where distances are measured parallel to the slope as seen in Fig. 2. In addition, with two sensors installed 30 cm apart at section 2, the velocity of the upper flow surface was derived using discrete correlation function of the two height signals of the two sensors. The mean front velocity at section 2 was back-calculated using data related to flow arrival time and distance at sections 1 and 2. The pressure applied by the flowing material on the small and large plates were also measured and a filtering mechanism was applied. The filtering of pressure

values was applied in order to remove oscillations caused by hard contacts due to large grains that impact the sensors. It was



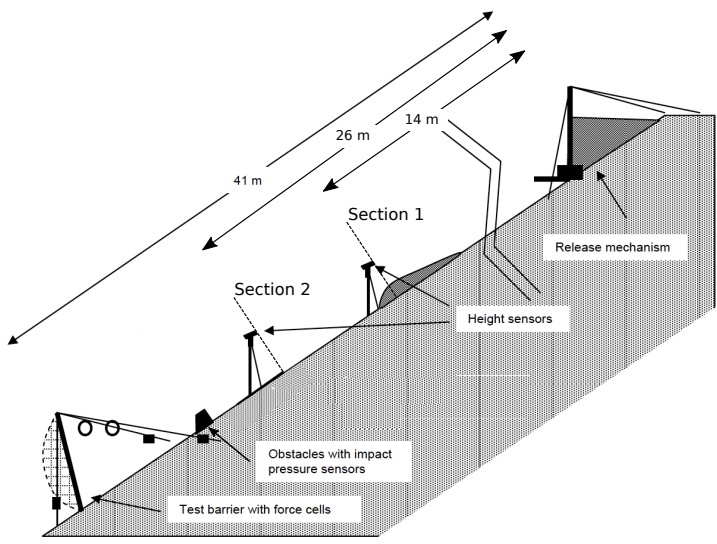

**Figure 2.** A schematic representation of the field experiment layout

applied by replacing each signal value by the mean value over an interval of 0.05 seconds. In Section 3.1, we investigated the possible relationships between those measured parameters (flow velocity and height) in the experiments and water and fines content of released materials. Previous studies of lab-scale experiments of hillslope debris flows showed that increasing water content had the largest positive change of the runout distance which might also indicate a possible increase in flowing velocity

(Hürlimann et al., 2015). In addition, a negative correlation was observed between clay content and runout distance. Both of those relations were found to be non-linear. Runout analysis is an important aspect of studying hillslope debris flows, but is out of the scope of this study, as it is not considered in the field experiment.

## 2.2 Discrete element simulations

The numerical simulation in this study was carried out using a Discrete Element Method (DEM). Nowadays DEM is widely
used for modeling granular media (Maurin et al., 2016; Papachristos et al., 2017; Mede et al., 2018). It is particularly efficient for static and dynamic simulation of granular assemblies where medium can be described at a microscopic scale. The method is based on an explicit finite difference scheme proposed by Cundall and Strack (1979). It applies for a collection of discrete bodies interacting with each other, governed by a contact law. Different contact forces can be considered both in the normal and the tangential direction. Calculations alternate between the application of Newton's second law to particles motion
and a force-displacement law for the particle interactions. In comparison with Finite Element Method (FEM), DEM makes large displacements between elements easy to simulate and computationally inexpensive, which is useful when dealing with discontinuous problems in granular medium.



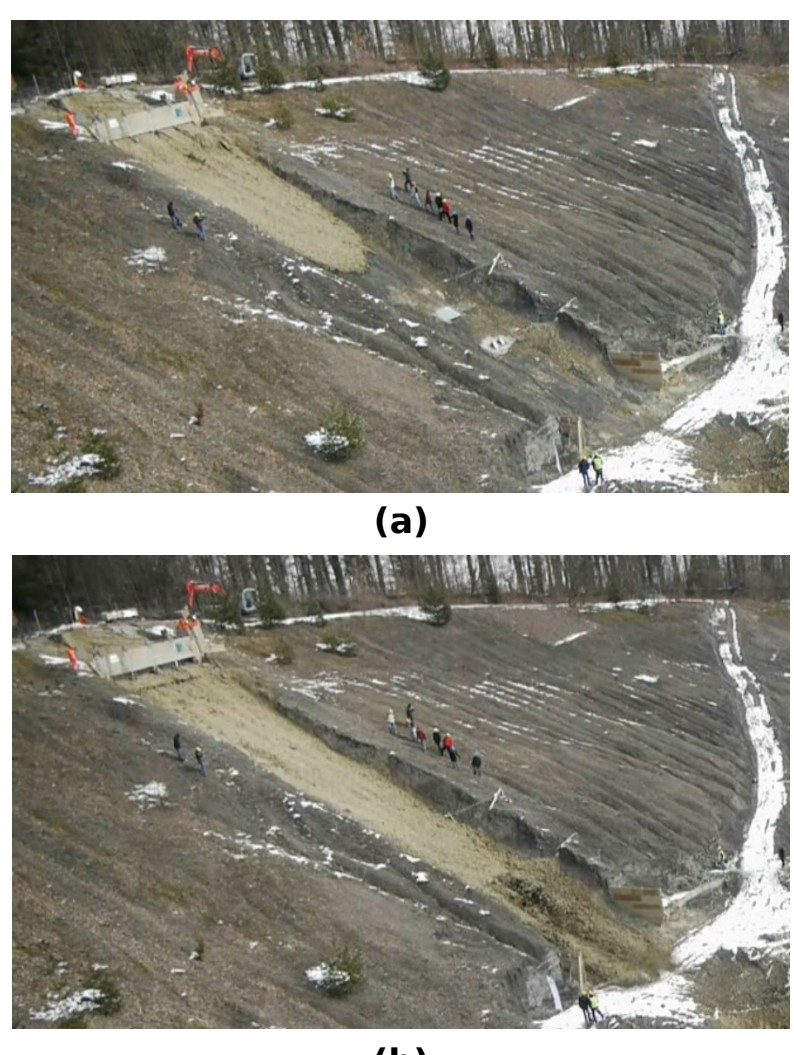

**Figure 3.** Screenshots of test 10: 3 seconds after release (a) 6 seconds after release (b)

YADE is an extensible open-source framework for DEM-based discrete numerical modeling (Šmilauer et al., 2010). The simulation loop in YADE starts with detecting contacts between particles. Next, the chosen contact law is applied, which results in new positions and velocities of the particles. YADE contains the main components for the application of DEM which include Newton's law, time integration algorithms, damping methods, collision detection, data classes (storing information about bodies and interactions) and command OpenGL methods for drawing popular geometries(Šmilauer et al., 2015).



### 2.2.1 Contact law

We imlemented an adhesive-elasto-plastic contact law in the YADE open source code based on the work of Luding (2008) in order to simulate the interaction between different particles of a flowing cohesive material. A hysteric force between two interacting particles $F_{hys}$ in the normal direction is calculated as follows (see also Fig. 4):

$$F_{hys} = \begin{cases} k_1 \delta & if\ k_2(\delta - \delta_0) \geqslant k_1 \delta \\ k_2(\delta - \delta_0) & if\ k_1(\delta) > k_2(\delta - \delta_0) > -k_c \delta \\ -k_c \delta & if\ -k_c \delta \geqslant k_2(\delta - \delta_0) \end{cases} \quad (1)$$

where $k_1$, $k_2$ and $k_c$ are stiffness parameters for loading, unloading and adhesive phases of contacts respectively. $\delta$ is the overlapping distance in the normal direction between the two particles. The stiffness parameters $k_1$, $k_2$ are related through the normal restitution coefficient ($\epsilon_n$) where $\epsilon_n = k_1/k_2$.

When a contact between two particles is established, with particles pushing into each other, the hysteric force would start
increasing linearly with the increase of the deformation $\delta$ along the path of $k_1$. The maximum reached deformation ($\delta_{max}$) will keep being updated as the deformation at the contact increases. Once unloading starts, the reached deformation would be temporarily saved as $\delta_{max}$ and the force-deformation path would be followed on the line indicated by the stiffness parameter $k_2$. In case of reloading, the path of $k_2$ would be followed again until reaching the maximum recorded deformation $\delta_{max}$ in which further loading would follow again the path of $k_1$. In case of unloading below $\delta_0$, which represents the interception
between $k_2$ path and the deformation axis and is calculated as $\delta_0 = (1 - k_1/k_2)\delta_{max}$, an adhesive force would be activated which is limited by the minimum force value $f_{min} = -k_c \delta_{min}$.

In addition to the hysteric force component, there is the classical viscous component of the force $f_{visc}$ (see for information Schwager and Poeschel 2007) which is the product of the viscous damping coefficient ($\gamma_n$), which depends on the chosen value of the restitution coefficient ($\epsilon_n$), and the velocity in the normal direction ($v_n$) which yields the following form of the
interaction force in the normal direction $\boldsymbol{F_n}$:

$$\boldsymbol{F_n} = (F_{hys} + \gamma_n v_n)\boldsymbol{n} \quad (2)$$

The tangential component of the normal force is governed by the classical Mohr-Coloumb failure criterion as follows:

$$\boldsymbol{F_t} = \begin{cases} \frac{k_t \boldsymbol{u_t}}{|k_t \boldsymbol{u_t}|}|\boldsymbol{F_n}|\tan\Phi_p & if |k_t \boldsymbol{u_t}| > |\boldsymbol{F_n}|\tan\Phi_p \\ k_t \boldsymbol{u_t} & otherwise \end{cases} \quad (3)$$

where $k_t$ is the tangential stiffness parameters, $u_t$ is the tangential displacement, $\Phi_p$ is the interparticle friction angle.
The normal stiffness of the contact between two particles ($k_1$) is calculated as (Catalano et al., 2014):

$$k_1 = \frac{2E_1 r_1 E_2 r_2}{E_1 r_1 + E_2 r_2} \quad (4)$$

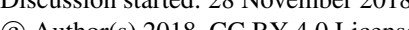



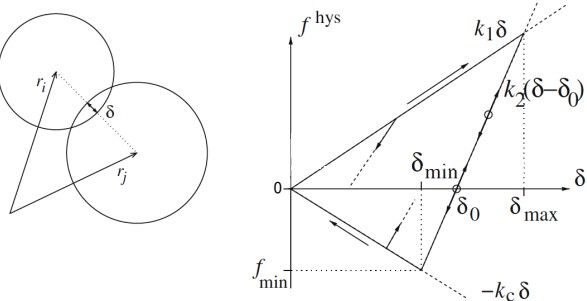

**Figure 4.** A schematic representation of the three phases of the contact law in normal direction

where $E_1$ and $E_2$ are the elastic moduli of the first and second particles respectively (both taken as $10^8$ Pa) and $r_1$ and $r_2$ are the radii of the first and second particles respectively.

### 2.2.2 Geometry and chosen parameters

In order to simulate the flow down an inclined chute, a wall object class was used in YADE to create a channel that is made of a base and two side walls. The chute was 41 m long, 8 m wide and channelized with 1 m height sides and inclined at $30°$. At the top of the channel, a rectangular reservoir was created where the flow starts. The reservoir was 7 m long, 8 m wide and 1.8 m high. Inside it, granular samples were created which were made of spherical particles with uniformly distributed diameters ranging from 50 mm to 100 mm and a mean diameter $d_{50} = 75$ mm. Particles had an intrinsic density $\rho_P = 2500 \ kg/m^{-3}$. Loose package of particles were generated at the beginning, with the total volume of each granular sample being around 50 $m^3$. Afterwards, under gravity deposition, particles got deposited at the bottom of the reservoir creating a dense and static sample. Once this step was achieved, the flow simulation starts by opening up part of the reservoir's gate upward for a distance of 0.8 m, thus allowing the particles to flow down the chute (Fig.5a). Different DEM tests were carried out varying the normal restitution coefficient ($\epsilon_n$) and the basal friction angle between particles and the chut's bottom ($\varphi_b$), while fixing the inclination angle $\alpha$ of the chute to $30°$. The interparticle friction $\varphi$ was $40°$. The tangential stiffness of the contact ($k_t$) was taken as $\frac{2}{7}k_n$ follwoing Silbert et al. (2001) and $k_c$ was equal to $k_1$. It is worth noting that the chosen geometrical configurations in DEM simulations corresponded to the those of the field experiment which were used to calibrate the model (Sec.2.1).

### 2.2.3 Mechanism of measuring parameters in YADE

Since YADE is a discrete element code, parameters such as height, velocity and pressure were characterized at the particle (micro) scale. In order to present those parameters in a macro-scale that is comparable with those of the experimental data, particle-scale parameters needed to be averaged in order to represent the flowing mass as a continuum medium. To compare the simulated mean front velocity with the experiment, the simulated flow arrival time was calculated at positions 1 and 2 and averaged over the distance between those two positions.





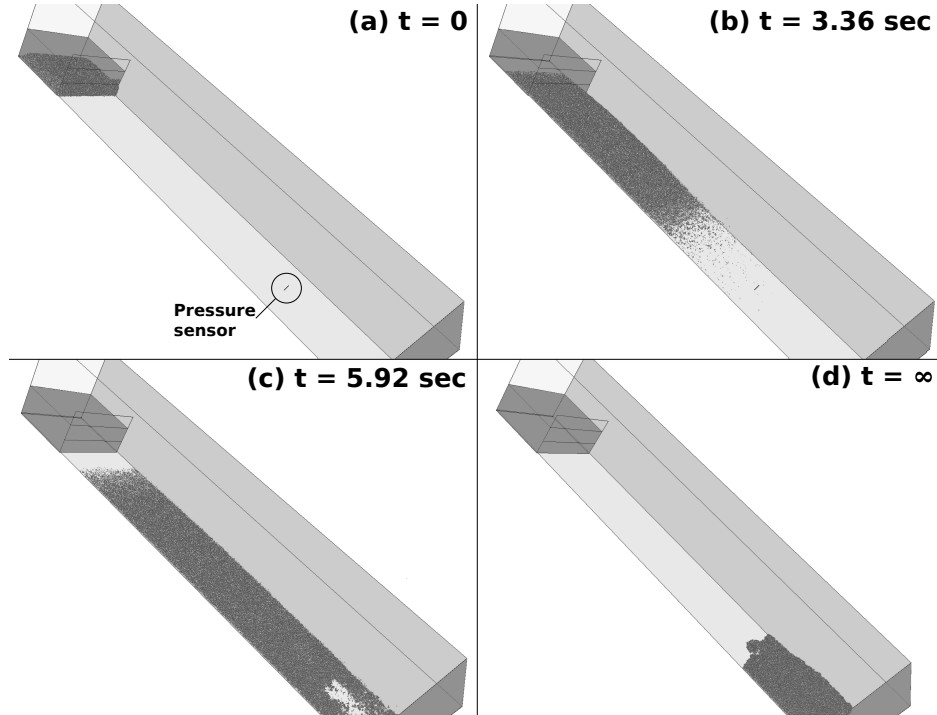

**Figure 5.** A series of screenshots during a DEM simulation with $\phi_b = 30°$ and $\epsilon_n = 0.3$: directly after opening the reservoir's gate (a), at t = 3.36 seconds (b), at t = 5.92 seconds (c) and at the end of the simulation (d).

The maximum flow height at position 2 is a value that represents the height of the main, coherent flowing body at that position. Firstly, a virtual box was used that was centered at position 2 and had a length 5 x $d_{50}$ and a width and height equal to those of the channel. Flow properties such as position coordinates in x (in the direction of the flow),y (traverse the flow) and z (perpendicular to the base) as well as flowing velocity in the direction of the flow were recorded each 0.1 second for all particles
within that box. Secondly, particles-height measurements at time periods between 25% and 75% of total impact duration were selected for each simulation, in order to exclude the disperse and dilute flow front and flow tail (Jiang and Towhata, 2013; Albaba et al., 2015), as the ones seen in Fig. 5b. Thirdly, the 90% cumulative frequency of flow height of particles within the box was selected. The maximum flow height in YADE that is compared to the experiment was then the maximum value of those selected cumulative frequencies for all the samples that were collected each 0.1 second.
For the impacting pressure, a rigid wall in YADE was installed at the same position where the large sensor was installed during the experiments and with the same dimensions (Fig.5a). Normal force $F_n$ applied to the wall by flowing particles was calculated as follows: $F_n = \sum_{i=1}^{n} F_{ni}$, where $F_{ni}$ is the normal force between a particle $i$ and the wall, and $n$ is the number of particles in contact with the wall at that moment. The pressure was then calculated as the ratio between applied force and the sensor's surface area. In order to be comparable with the experiment, the same filtering interval (0.05 seconds) was applied to
the pressure signal in the DEM simulations.





**Table 1.** Main flow characteristics and measured parameters for experimental tests selected for model calibration

| Release No. | Wet density $(kg/m^3)$ | Water mass fraction (%) | Fine mass fraction (%) | Max. flow height at pos. 2 (m) | Mean front velocity (m/s) | Max. pressure on large sensor (kPa) |
|---|---|---|---|---|---|---|
| 9 | 1,790 | 28 | 48 | 0.29 | 10.2 | 65.9 |
| 10 | 1,900 | 18 | 21 | 0.4 | 8.2 | 96 |
| 11 | 2,060 | 16 | 27 | 0.38 | 9 | 94.6 |
| 13 | 1,880 | 22 | 28 | 0.33 | 8.4 | 98.5 |
| 14 | 1,990 | 17 | 25 | 0.4 | 9.1 | 138 |
| 15 | 1,830 | 23 | 25 | 0.37 | 8.9 | 109.4 |
| 16 | 2,110 | 14 | 41 | 0.37 | 6.4 | 69.2 |

## 2.3 Comparison of DEM and experimental data

To calibrate the DEM model, only first releases of selected tests from the field experiment were considered in order to avoid the possible disturbance of measured parameters in the experiments due to the presence of deposits of previous releases (tests abbreviated as X.1 in Table 4 in Bugnion et al. (2012) where X is the test number). In the current DEM model, no material deposited on the inclined plane and thus the multiple releases would be difficult to reproduce. Seven tests from the experimental data were selected to be compared with the DEM model results. They are numbered with the same digits as in Bugnion et al. (2012). Table 1 summarizes the main material properties of these tests and the measured parameters.

A series of simulations varying the parameter set $(\phi_b, \epsilon_n)$ were carried out. A range between 20-40° was selcted for $\phi_b$ with a step-wise increase of 5° while $\epsilon_n$ was varied between 0.3-0.45 with an increment of 0.03. Those ranges were selected based on preliminary model tests. Simulations with $\phi_b$ and $\epsilon_n$ values that are outside the selected ranges were found to either result in very fast flows with very high impacting pressures or flows that would not slide along the channel. Afterwards, all carried out simulations were compared with each selected experiment in order to find the best-fit in terms of: maximum flow height at position 2 ($H_{max}$), mean front velocity between positions 1 and 2 ($V_{mean}$), and maximum applied pressure to the large sensor ($P_{max}$). Results of pressures on smaller sensors were ignored because they were not measured for each test in Bugnion et al. (2012). In addition, in DEM simulations, the smaller the sensor, the more discrete in nature the force signal would be due to the presence of fewer particles per impact. A best-fit for each selected experimental test was determined as the lowest percentage ($R_{min}$) of error for the three parameters as follows:

$$R_{min} = \min_{\forall i \in n_s} \left( \frac{\sqrt{(H_{DEM})_i - H_{EXP}}}{H_{EXP}} + \frac{\sqrt{(V_{DEM})_i - V_{EXP}}}{V_{EXP}} + \frac{\sqrt{(P_{DEM})_i - P_{EXP}}}{P_{EXP}} \right) \qquad (5)$$

where $n_s$ is the number of simulations

After finding the best parameter set $(\phi_b, \epsilon_n)$ for each experiment, possible relationships between these sets of parameters and initial condition of the granular samples (i.e. water and fine content) were investigated.





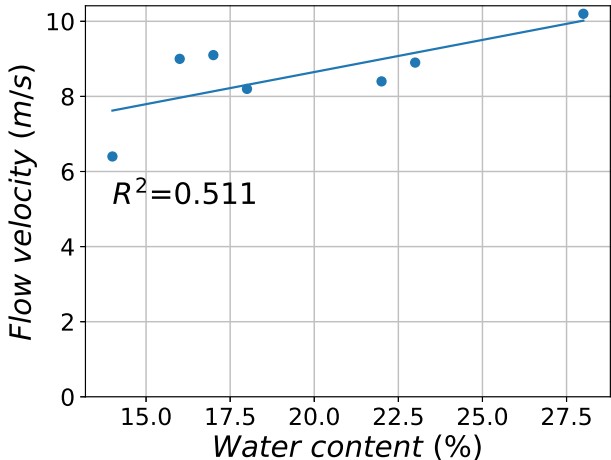

**Figure 6.** Relation between the water content of the granular material of selected experimental tests and the observed mean front velocity

## 2.4 Sensitivity analysis of DEM parameters

A sensitivity analysis study was carried out in order to investigate the effect of change of different parameters on the model's results concerning maximum flow height, mean flow front velocity and maximum applied pressure on the large plate. The effect of variation of each of the following parameters: $\phi_b$, $\epsilon_n$, $k_1$, $k_c$, $d_{50}$ and $\alpha$ was investigated in detail. To carry out these
sensitivity analyses, we selected a best-fit parameter set out of all simulated sets.

## 3 Results

### 3.1 Analysis of experimental data

For the seven selected experimental tests, a relationship seems to exist between the water content in the granular material prepared in the reservoir and the recorded mean front velocity between sections 1 and 2 in the field experiment (Fig.6). For
example, test no. 16 had a water content of 14% and its recorded mean front velocity was found to be 6.4 m/s. In addition, water contents between 16-23% had similar recorded front velocities ranging from 8.2 to 9.1 m/s for tests number 10 to 15. Test no. 9, which had the richest water content, had the highest mean front velocity of 10.2 m/s. This observed increase in flow velocity with increasing water content could be due to the decrease of basal friction with the flowing material which leads to higher flowing velocities. However, a best-fit of the results might be better represented with a non-linear equation in comparison with
the regression line presented in Fig. 6, which is similar to observations of Hürlimann et al. (2015).

Another observation can be made regarding the relationship between the amount of fines content (silt and clay) in the prepared material in the reservoir and the maximum flow height recorded at position 2, which is found to be inversely proportional (Fig.7). At low levels of fines content (21-28%), maximum flow heights were found to be between 0.33 and 0.40 meters. A





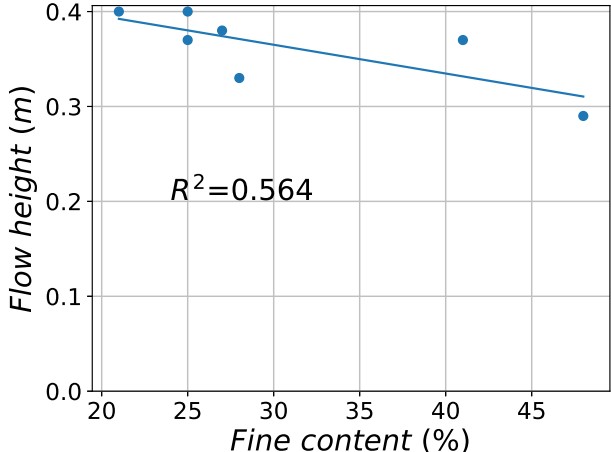

**Figure 7.** Relation between the fine content (silt and clay) of the granular material of selected experimental tests and the observed maximum flow height

high increase of fines content, like in test no. 9, resulted in a drop in measured flow height to 0.29 meters. However, such a relationship is not very evidently inverse. This is because for test no. 16, although the fines content was high (41%), the recorded flow height was found to be larger than the average for all tests (0.37 m).

In general, such observations of possible relations between flow initial conditions of the granular material and flowing

height and velocity supports a calibration process based on those two parameters in addition to the pressure applied to the large sensor. These three parameters were found to vary with each test, but with different percentages. Figure 8 shows the variation around the mean of each parameter for the seven selected tests. Maximum flow height at position 2 is found to have the lowest variation around the mean in comparison with other parameters. The maximum variation is recorded for test no. 9 which has a flow height of 0.29 m which is 20% lower than the mean. Mean front velocity between sections 1 and 2 is found to have higher

oscillations around the mean for different tests. Velocity of test no. 16 is 6.4 m/s which is 22% lower than the mean of all tests. The highest variation is present in the pressure values which varied between 15% and 42% below and above the mean for tests 9 and 14 respectively.

### 3.2    Cross comparison between DEM simulations and field experiments

#### 3.2.1    Flow height and velocity

Figure 9 shows the comparison between the measured maximum flow height at section 2 in the selected field experiments with values of their corresponding best-fit numerical simulations. It can be seen that a very good agreement is observed for tests 10, 11 and 15 when compared with the DEM results. For tests 13 and 16, a relatively good agreement is observed with the maximum margin of error being 15%. The least agreement is shown for test no. 9 which has an error of almost +50%. This test showed the highest variation from the mean when compared with other experimental tests (Fig.8).



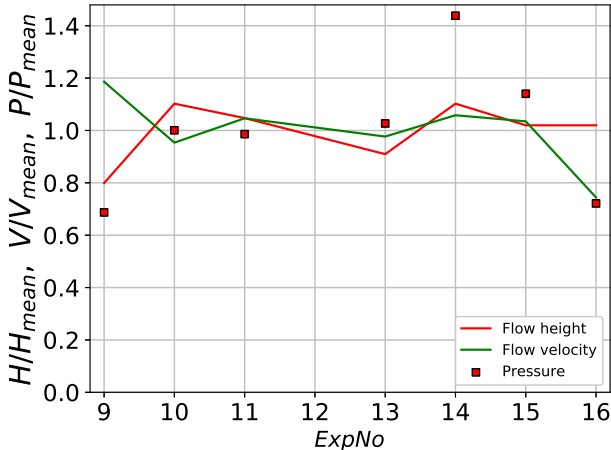

**Figure 8.** Fluctuation of measured parameters in the experiments around the mean for flow height, flow velocity and impact pressure

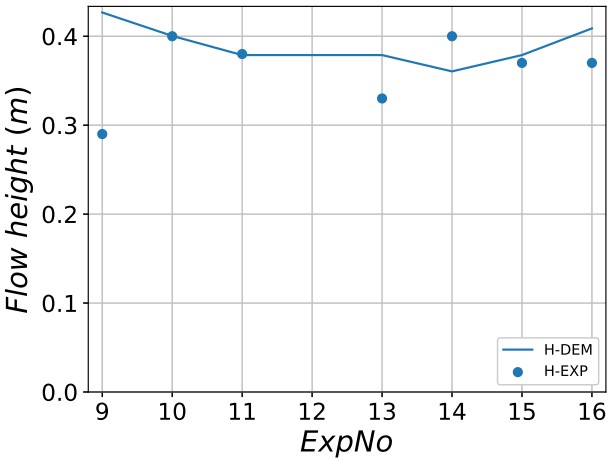

**Figure 9.** Maximum flow height at section 2 for experimental data (H-Exp) and their corresponding best-fit DEM simulations (H-DEM)

For flowing velocity, a better general agreement between the experiment and DEM results is observed (Fig.10). For example, for test no. 9, the maximum observed front velocity is reproduced by the DEM model. For tests 10 to 15, a relatively similar flow velocity of 8-9 m/s is observed for both the experiment and the model. The least agreement is observed for test no. 16 for the which the experimental value is 6.4 m/s and the corresponding best-fit simulation flow velocity is 8.5 m/s.

### 5  3.2.2  Impact pressure

Figure 11 shows the maximum pressure applied to the large sensor for both the experiment and DEM simulations. Very well agreement is observed for most experiments when compared with their corresponding best-fit simulations. For test no. 9, the





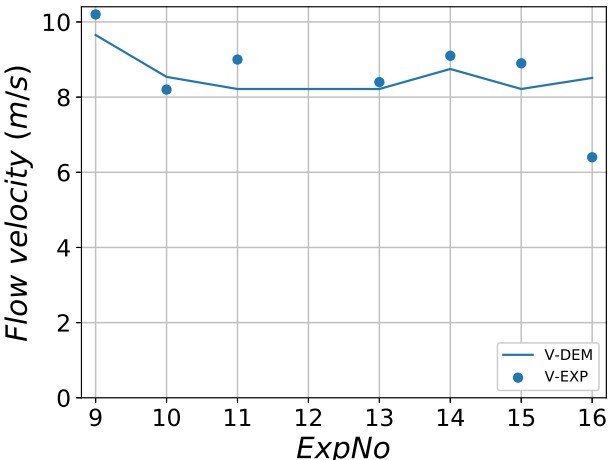

**Figure 10.** Flow front velocity, measured between sections 1 and 2, for experimental data and their corresponding best-fit DEM simulations

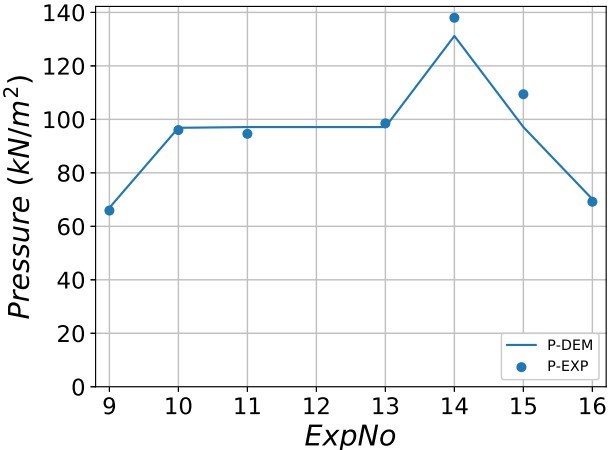

**Figure 11.** Maximum pressure applied to the large sensor for experimental data and their corresponding best-fit DEM simulations

maximum pressure recorded during the experiment was equal to 65.9 kPa while the corresponding best-fit value recorded in the simulation was 66.9 kPa, resulting in an error of +1.5%. Tests 10, 11, 13 and 16 had similarly low values of error when comparing pressures between the experiments and best-fit simulations. The least agreement, is observed for tests no. 14 and 15 where the errors are 4.9% and 11.2% respectively.

5     It is however important to compare the pressure evolution for the different tests, in addition to the comparison with peak pressure values. This is because the same peak pressure value could be achieved with different pressure evolutions. A filtering window of 0.2 seconds has been applied to both pressure signals of the experiment and the DEM simulations in order to obtain pressure evolution curves that can be compared properly.





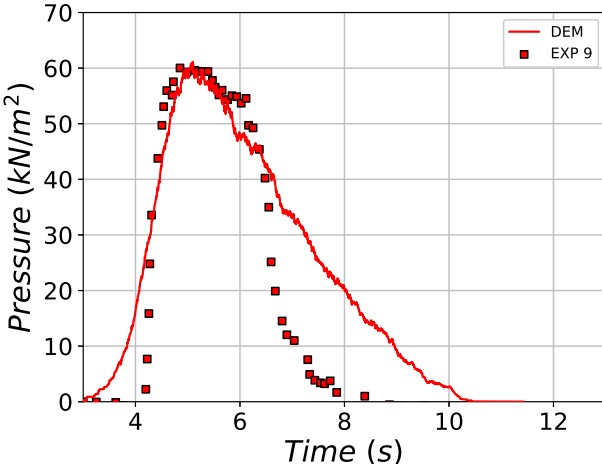

**Figure 12.** Evolution of pressure on the large plate for test No. 9 and its respective DEM best-fit simulation

The evolution of pressure applied to the large sensor during the experimental test no. 9 is shown in Fig. 12 along with its corresponding best-fit DEM simulation. At the beginning of the impact (3 < t < 4.2 seconds), the DEM curve starts recording pressure values which are due to the dilute group of particles that are detached from the main flow and individually impact the rigid wall representing the pressure sensor in the simulation. Afterwards, the two curves agree well with each other until

reaching similar peak values at similar time points (60 and 61 kPa for the experiment and DEM respectively). After the peak (around t = 5.2 seconds), both measured pressures start decreasing with similar rates until t = 6.35 seconds. Further decrease in pressure is found to be faster in the experiment in comparison with DEM where the decrease occurs over longer periods of time. At the end, pressure signal in DEM is found to lag 2 seconds behind that of the experiment.

Similar observations can be made when comparing pressure values of test no. 14 with its corresponding DEM simulation

(Fig.13). A first phase of impact of the dilute group of particles causes pressure values to increase for the DEM with no equivalent increase in the experiment. Afterwards, both pressure curves agree well until reaching very similar peaks (113.5 and 114.7 kPa for the experiment and DEM respectively). The decrease in pressure that follows the reached peaks has similar rates for both the experiment and DEM until t = 8 seconds. Afterwards, pressure values of the experiment decrease faster while those of DEM model lag behind causing the total impact duration of the DEM to be larger than the one of test no. 14.

Tests 11, 13 and 15, which had very similar initial conditions of the granular material and also similar values of measured parameters (height velocity and pressure), were found to be best fitted with the same parameter set in YADE ($\epsilon_n = 0.3, \phi_b = 30°$). Comparison of pressure values recorded in the three experiments and corresponding DEM simulation reveal similar trends (Fig. 14). At the beginning, the evolution of pressure in DEM starts effectively at t = 4 seconds, less than one second earlier than those of the experiments. Afterwards, pressure signal of test no. 13 starts slightly earlier than those of tests no.

11 and 15. Higher values of pressures are recorded as more flow accumulates behind the pressure sensor until reaching the peak. The reached peaks are very similar in values when comparing the DEM simulation with the different tests (86.3, 89.6, 85





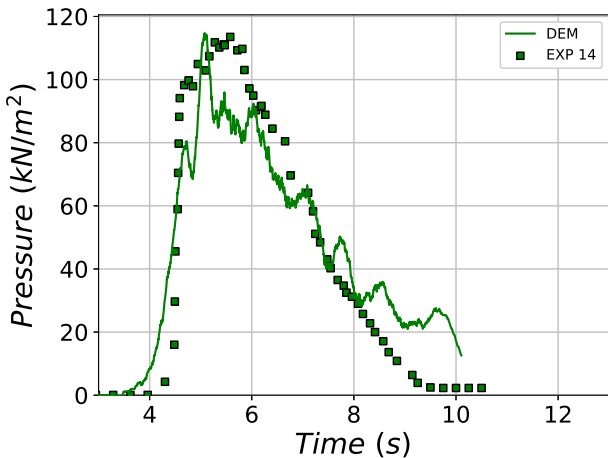

**Figure 13.** Evolution of pressure on the large plate for test No. 14 and its respective DEM best-fit simulation

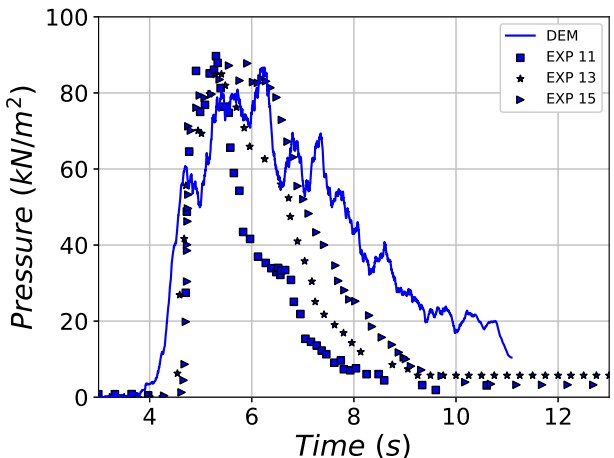

**Figure 14.** Evolution of pressure on the large plate for test No. 11,13 and 15 and their respective DEM best-fit simulation

and 87.7 kPa for DEM, test 11, test 13 and test 15 respectively), although the recorded peak of the DEM simulation is delayed compared to those of the experimental tests. After the peak, pressures values of test no. 11 are found to decrease faster followed by test no. 13. Those of tests no. 15 are found to lag less than one second behind. DEM pressure values are the latest to start decreasing, resulting in a longer impact duration.

5      The last comparison concerns test no. 16 which is found to agree well with its corresponding best-fit DEM simulation concerning pressure evolution. Apart from the early start of the DEM curve, which is due to the dilute front, both curves are found to start at very similar times (around t = 4.3 seconds). Afterwards, peak pressure values are rapidly reached for both DEM simulation (55 kPa) and the experimental test (62 kPa), which might indicate the presence of coarse-grained particles in





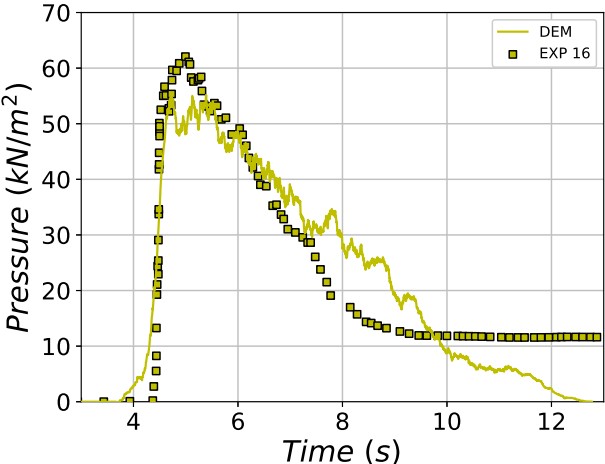

**Figure 15.** Evolution of pressure on the large plate for test No. 16 and its respective DEM best-fit simulation

the frontal part of the flow. Then both pressure curves start decreasing with similar rates until t = 6.4 seconds. Pressure values of test no. 16 are then found to decrease faster than those of the simulation until reaching static pressure values of around 12 kPa, which indicates the deposit of some material on the pressure sensor. The DEM curve progressively decreases over a longer period of time until decaying to zero at t = 12.5 seconds.

### 3.2.3 Best-fit

Results of all DEM simulations are best fitted against experimental data of tests in Veltheim site using Eq. (5). Figure 16 shows the correspondence between experimental test and their respective best-fit DEM parameters, based on comparisons of flow height, flow velocity and impact pressure on the large sensor. Tests 10, 11, 13, 14 and 15 are found to be reproducible with very similar values of parameter set $\epsilon_n$ and $\phi_b$ (zone limited by red dashed lines in Fig. 16). Those tests are found to have similar values of water content in the granular material prepared in the reservoir (between 16 and 23%). In addition, they tend to have similar values of fine content (silt and clay) which ranges between 21 and 28 %. Test no. 9 is best reproduced by $\epsilon_n = 0.45$ and $\phi_b = 25°$ while test no. 16 is best reproduced by $\epsilon_n = 0.36$ and $\phi_b = 40°$. Further analysis of the best-fit results and possible relations between model parameters and granular sample's initial conditions are discussed in Section 4.2.

### 3.3 Results of Sensitivity Analysis

In this section, a detailed parametric study is carried out for the set of parameter of the simulation which was found to agree the most with the different experimental tests (i.e. simulation with $\phi_b$=30 deg and $\epsilon_n$=0.3). The effects of variation of: $\phi_b$, $\epsilon_n$, $k_c$, $d_{50}$ and the chute inclination angle ($\alpha$) are introduced. The observed effects on the measured flow height, flow velocity and applied pressure are then discussed. For convenience, height, velocity and pressure results of the different sensitivity analysis tests are normalized by values of the baseline simulation with $\phi_b$=30° and $\epsilon_n$=0.3.





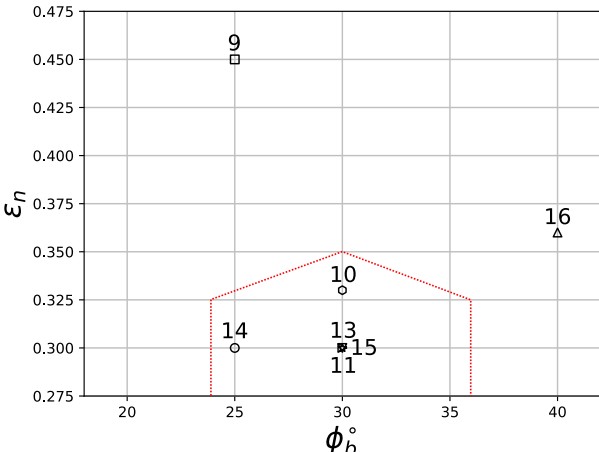

**Figure 16.** The best fitting set of parameters ($\epsilon_n$ *and* $\phi_b$) for each of the selected experimental tests based on comparison of flow height, mean front velocity and maximum applied pressure

### 3.3.1 Mean particle diameter

Four samples with different values of $d_{50}$ are tested: 75, 100, 125 and 150 mm. Values of the normalized maximum flow height at position 2, mean front velocity and maximum applied pressure on the large plate are shown in Fig. 17. The maximum flow height at position 2 is found to increase when moving from $d_{50}$ = 75 mm to 100 mm while a smaller increase is observed for

further increase of mean diameter (simulations with $d_{50}$ = 125 and 150 mm). This increase is due to the presence of larger particles in the virtual box where flow height is measured in the model perpendicular to the base of the chute. However, all in all, the observed difference in flow height for the different mean diameters is rather limited and is found not to exceed 25% although the mean diameter is doubled. Regarding the flow velocity, negligible changes are observed when varying the mean diameter. Flows in the different simulations are found to arrive at similar times at position 2 thus having similar mean

front velocities. Concerning the impact pressure on the large plate, a considerable difference is observed when comparing the different particle sizes. The maximum pressure for simulation with $d_{50}$ = 150 mm is 1.64 times larger than that of $d_{50}$ = 75 mm. In addition, the pressure is found to increase with the increase of mean diameter, except for $d_{50}$ = 100 and 125 mm. For those two tests, normalized pressure values are 1.30 and 1.18 respectively. These observations support the assumption that pressure values are mostly dominated by hard contacts with large solid grains. Such contacts influence the pressure signal

although their influence is reduced by the application of filtering windows. The probability of a large particle impacting the pressure sensor increases by increasing the mean diameter because of the decrease in the number of particles (as the volume of the different tests is fixed at 50 $m^3$). As particles grow in size and reduce in number, impact mechanisms tend to be similar to those of rockfalls (points loads) rather than those debris flows (gradual cross-sectionally-spread loads). The decrease in normalized pressure of the test with $d_{50}$ = 125 mm in comparison with that of $d_{50}$ = 100 mm can be understood through the

random positioning of particles in the sampling box upstream the channel. Particles are created with random position at each





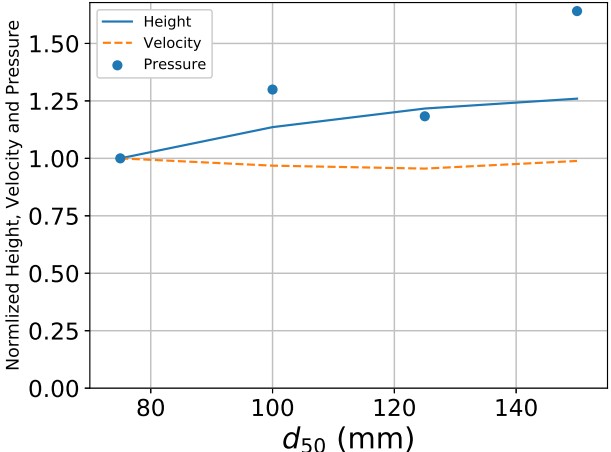

**Figure 17.** Variation of flow height, velocity and pressure for different values of mean particle diameter ($d_{50}$) when normalized by results of the simulation with $\phi_b$=30°, $\epsilon_n$=0.3, $d_{50}$=75 mm and $\alpha$ = 30°

simulation which leads to different distributions of particles in flowing phase. This finding supports the importance of filtering out pressure signals that are due to individual contacts with big particles.

### 3.3.2 Normal restitution coefficient ($\epsilon_n$)

Figure 18 shows the effect of varying the normal restitution coefficient ($\epsilon_n$) on the flow height, the velocity and the pressure.

Six values of $\epsilon_n$ are tested: 0.30, 0.33, 0.36, 0.39, 0.42 and 0.45. Modifying the value of $\epsilon_n$ results in a change in $k_2$ since $k_1$ is fixed and $\epsilon_n = k_1/k_2$. For the flowing height and velocity, a small gradual increase is observed in their values with the increase in $\epsilon_n$. This might be due to the decrease in plastic deformation at the microscale, which leads to more dispersion of particles away from the center of the flowing mass. On the contrary, for the impact pressure, an increase in $\epsilon_n$ results in a considerable decrease in the value of maximum impact pressure. The difference is very small for an increase of $\epsilon_n$ from 0.3 to

0.33. However, further increase of $\epsilon_n$ results in a systematic decrease of the maximum applied pressure, which reaches a 40% decrease for $\epsilon_n$ = 0.45 in comparison to $\epsilon_n$=0.3. This decrease could be due to a large dispersion of particles leading to a more gradual impact on the sensor.

### 3.3.3 Basal friction coefficient

The basal friction angle ($\phi_b$) of the chute is varied from 20 to and 40° with a 5° increment. Figure 19 shows the effect of

this variation on measured parameters in the simulations. Flow height is found to increase when increasing the basal friction. Simulations with $\phi_b$=20° record maximum flow heights that are 32% lower than those with $\phi_b$=40°. This could be due to the increase in the basal resistance to flow shearing which increases the number of particles in the vertical direction perpendicular to chute base. An inverse relation is observed for the flow velocity which is found to decrease by increasing friction angle.





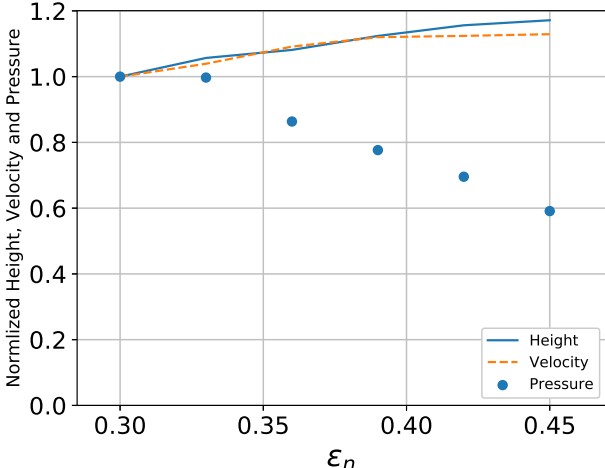

**Figure 18.** Variation of flow height, velocity and pressure for different values of normal restitution coefficient ($\epsilon_n$) when normalized by results of the simulation with $\phi_b$=30°, $\epsilon_n$=0.3, $d_{50}$=75 mm and $\alpha$ = 30°

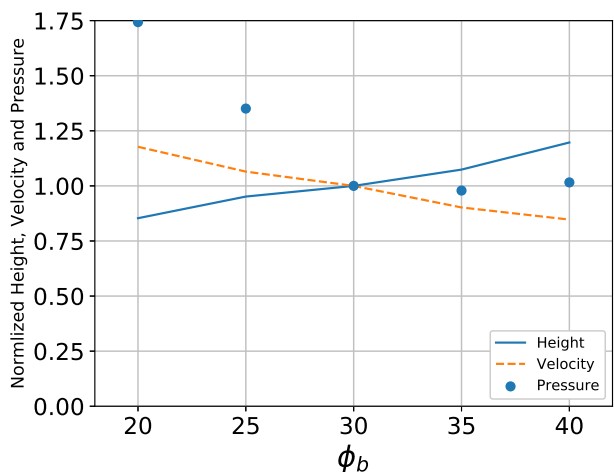

**Figure 19.** Variation of flow height, velocity and pressure for different values of basal friction angle ($\phi_b$) when normalized by results of the simulation with $\phi_b$=30°, $\epsilon_n$=0.3, $d_{50}$=75 mm and $\alpha$ = 30°

This is due to the increase in the resistance to movement by the chute base. This decrease in flowing velocity has a direct impact on the value of maximum impact pressure, which is found to decrease with increasing basal friction. A sharp decrease is observed for pressure values when decreasing basal friction angle from 20 to 30°. Further decrease is however found to have a limited effect on the recorded pressure values. Overall, the observed relationship between flowing velocity and impact pressure is governed by the increase in kinetic energy of the flowing mass (Faug et al., 2009; Jiang and Towhata, 2013; Albaba et al., 2018).




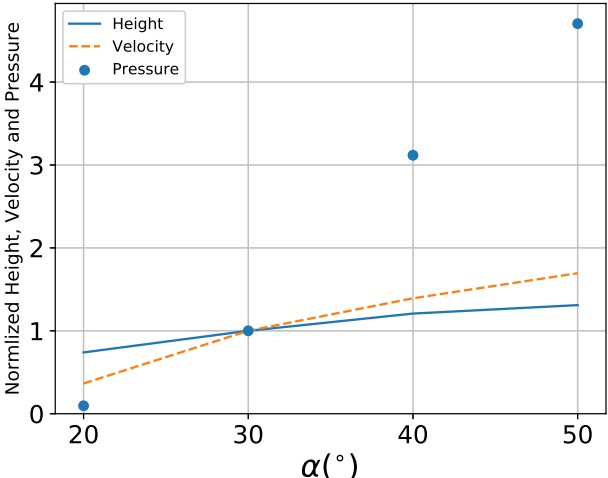

**Figure 20.** Variation of flow height, velocity and pressure for different values of inclination angle ($\alpha$) when normalized by results of the simulation with $\phi_b$=30°, $\epsilon_n$=0.3, $d_{50}$=75 mm and $\alpha$ = 30°

### 3.3.4 Inclination angle

To investigate the effect of changing the chute inclination angle ($\alpha$), which was fixed at 30° for all other simulations, corresponding to the test site in Veltheim, three additional values of $\alpha$ are tested: 20, 40 and 50°. It was noted during the simulations that an inclination angle of 20° did not reproduce a dense flow but a very discrete flow of particles instead. This is because the value of basal friction $\phi_b$ is larger than $\alpha$ (GDR-MiDi, 2004). As a result, the simulation case with $\phi_b$ = 20° will be ignored in the results analysis.

The change in inclination angle has an effect on the maximum flow height at position 2 (Fig. 19), which increases with 25% for a change in $\alpha$ from 30 to 50 °. A larger effect is observed for the mean front velocity which increases by 65% for an increase in $\alpha$ to 50°. Such increase has a direct link to the maximum applied pressure which increased by 470%.

## 4  Discussion of obtained results

### 4.1  Phenomenology of impacting pressure

Results of the field experiments showed that the highest varying parameter was the maximum impact pressure (Fig. 8). These high variations in pressure were possibly due to the interaction between large boulders of the flow and measuring senors in short periods of time, although this effect was minimized by filtering the data over a period of 0.05 seconds. This phenomenon is supported by the fact that although some tests had similar initial conditions (water and fines content) and also similar flowing height and velocity, the maximum recorded pressure was largely different. This is clear when comparing tests no. 11 and 14 which had similar values of initial and flowing conditions but different pressures of 94.6 kPa and 138 kPa respectively.



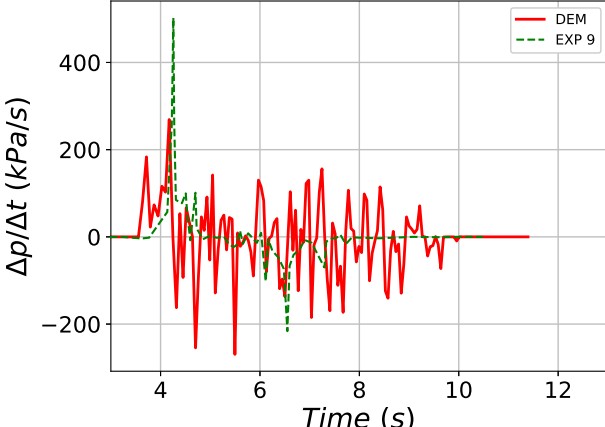

**Figure 21.** Rate of change of pressure applied to the large sensor during test No. 9 and its respective DEM best-fit simulation

Comparisons of pressure evolution of both the selected experimental tests and the DEM simulations revealed general agreements in peak pressure values and impacting trend. However, some discrepancies were observed concerning impact rate and duration. The nature of DEM simulations, in which a group of particles interact at the microscale, might have contributed to these observed differences. For example, pressure signals during DEM simulations were found to start earlier than those of the

experimental tests. This was due to the detachment of a group of particles from the mean part of the flow, causing particles to form a dilute frontal part which impacts the pressure sensor early (Fig.5b). Furthermore, the decreasing phase of the pressure signal was found to last longer for DEM simulations in comparison with experimental tests. The formation of a dilute tail could be responsible for that as it needed longer time to fully interact with the sensor (a compression phase of the dilute part needs first to occur).

Another difference between the experiments and the simulations is the rate of change in pressure signal ($\Delta p/\Delta t$). Faster loading/unloading phases are generally observed for the experimental tests than for those of the simulations. This is clear when comparing the rate of change between the two curves for test no. 9 (Fig. 21). For the experimental curve, a sudden sharp increase is observed in the pressure rate at t = 4.25 seconds due to fast increase in pressure in very short period of time. Afterwards, a quasi steady-state of pressure rate is observed as pressure reaches its maximum value. A second sharp peak,

which is negative, is observed at around t = 6.5 seconds as the flow mass starts to vanish and more impact on the sensor exist. On the other hand, impact rates of the numerical model are found to have lower local peaks during the impact duration which lasts longer than that of the experiment. In contrast to the experiment, the successive local pressure peaks could indicate an impact mechanism based on a successive surges which in turn could explain the longer duration of impact needed for numerical simulations to end.

In addition to impact duration and rate, the fluctuating nature of the raw pressure signal causes the definition of peak pressure value to be dependent on the sensor's frequency and on the filtering window of the raw data. Figure 22 shows the signal of pressure history for test no. 15. It can be observed that, for shorter periods of time, extreme values of peak pressure





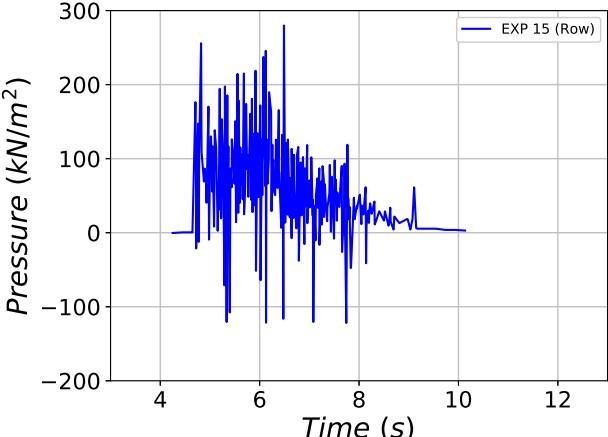

**Figure 22.** Row signal of the evolution of pressure on the large plate for test No. 15

are recorded (up to 280 kPa). These are possibly due to the contact at the local scale with large solid particles of the flow. Similar fluctuations of pressure (or force) curves simulated by DEM models have been reported frequently (Faug et al., 2009; Kneib et al., 2017; Shen et al., 2018b). For the sake of model calibration using experimental data, these extreme signals were minimized by applying a filtering window. However, for practitioners who are interested in houses/infrastructures resistance to

hillslope debris flows, taking into account such impulses might be important in order to avoid failures of structures at the local scale due to the interaction with large solid particles within the flowing material.

## 4.2  Best-fit parameter set

Results of best-fit simulations based on flow height, mean front velocity and maximum applied pressure revealed that experimental tests (11, 13 and 15) could be reproduced using the same parameter set in YADE (i.e. same $\epsilon_n$ and $\phi_b$) as already seen

in Section 3.2.3. Test no. 9 and 16 were found to be reproduced with very different parameter set. The difference in numerical best-fit parameters can not be explained by the pressure values which are very similar for both tests (65.9 and 69.2 KPa for tests 9 and 16 receptively). Despite this fact, test no. 9 recorded the highest mean front velocity (10.2 m/s), possibly due to having the highest water content (28%). Unlike other tests, such high speed flowing material did not contribute to a high pressure value mainly because this test had the highest fine material (i.e. lowest percentage of sand) and consequently the lowest wet density

(1790 $kg/m^3$). All these conditions are reflected in the value of best-fit basal friction angle parameter in the DEM simulation, which is lowest among all best-fit simulations.

On the other hand, test no. 16 had a low water content in the released material (14%) but a relatively high fine content (41%). This contributed significantly to its wet density making it the highest among all tests (2110 $kg/m^3$). However, the low water content of the granular material might have led to its low mean front velocity, which was the lowest among all tests (6.4 m/s).

This low flowing velocity was probably the main reason for the low maximum pressure value recorded during that test. All



these initial conditions (especially the low water content) were reflected in the value of best-fit basal friction angle parameter in the DEM simulation, which is highest among all best-fit simulations ($40°$).

It is worth noting that attempts to base the best-fit solely on one part of Eq. (5) did not produce consistent results in terms of the relation between the initial conditions of the experimental test and the model parameter set $\epsilon_n$ and $\phi_b$ (See Fig. A1 in Appendix A which shows results of best-fit comparisons based separately on either the flow height, the flow velocity or the impact pressure).

All in all, to draw strong conclusions on the relationship between initial conditions of granular samples and YADE model parameters, experimental data with a wider range of both water content and fine content are needed. The experimental data considered here had a narrow range of variation of both of those parameters, which was reflected in the narrow variations of maximum flow height values. Since field experiments are expensive and difficult to organize, lab experiments can be used instead in order to study the effect of those parameters in detail.

## 5   Conclusions

Rapid urbanization of mountainous areas contributed to the focus on studying the different types of mass movement such as landslides and hillslope debris flows. In this study, a Discrete-element-based contact law was implemented for the purpose of modeling hillsope debris flow. The model has three phases which are elastic, plastic and adhesive phase. The model capabilities in reproducing filed-scale hillslope debris flow experiments were tested in detail. A group of seven experimental of tests were selected with varying levels of bulk density, water content and fine content. In each experiment, maximum flow height at a defined section, mean front velocity and maximum impact pressure applied to a measuring sensor were measured. 30 numerical simulations were carried out by varying two parameters in the numerical model (basal friction angle $\phi_b$ and normal restitution coefficient $\epsilon_n$). Calibration of the model against experimental data was based on finding the best-fit set of parameters $\phi_b$ and $\epsilon_n$ of the model that matches each selected experiment concerning flow height, mean front velocity and applied pressure.

We conclude that there is a very good agreement between the model and experiments was observed concerning mean front velocity and maximum applied pressure, with lesser agreement of flow height. Detailed comparisons of pressure evolution between different selected experiments and simulations revealed the model's capability of reproducing observed pressure curves, especially during the primary loading phase leading to maximum pressure. However, since the model did not simulate the deposition of material on the inclined channel, a post-peak unloading phase similar to the experiments could not be reproduced. The analysis of the best-fit between the model and the experiments showed that three experimental tests were best reproduced with the same $\phi_b$ and $\epsilon_n$ parameter combination. These experiments were found to share similar medium values of water and fine content. Increasing the basal friction in the model led to simulations matching the experiment with lowest water content and highest bulk density. On the contrary, a higher value of $\epsilon_n$ and relatively low value of $\phi_b$ were needed to reproduce the test with the highest water content and the lowest bulk density. All these findings suggest that a link exists between the model parameters and initial conditions of granular samples. Such a link should be further investigated in detail on the basis of additional hillslope debris flow experiments.



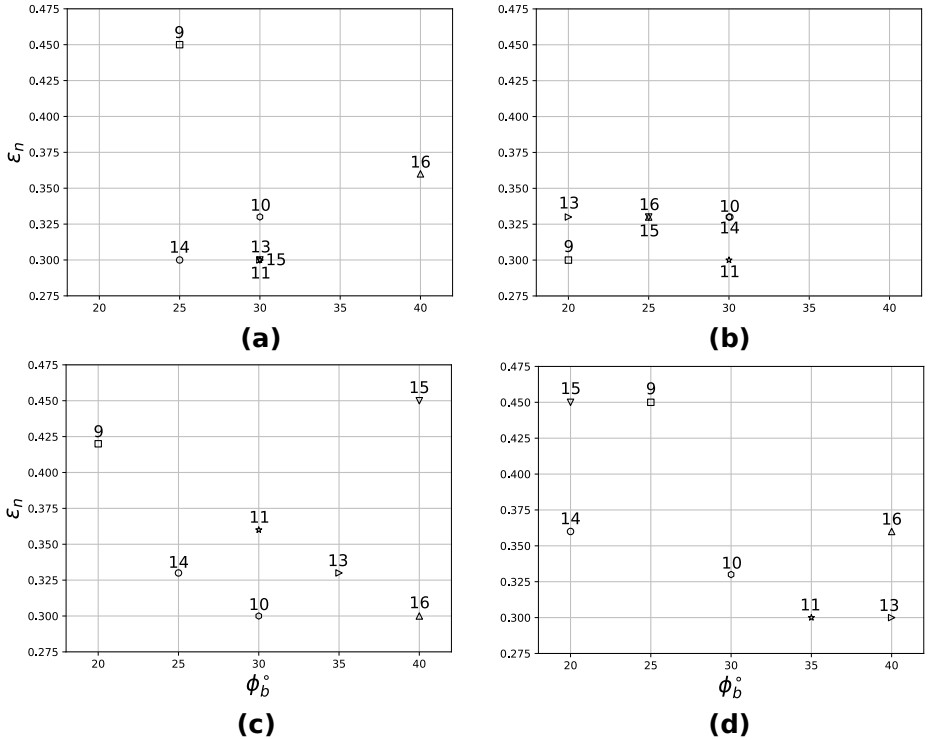

**Figure A1.** The best fitting set of parameters ($\epsilon_n \ and \ \phi_b$) for each of the selected experimental tests based on: all three parameters (a), only the flow maximum height (b), only the mean front velocity (c) and only the maximum impact pressure (d)

### Appendix A: Model calibration based separately on the flow height, velocity and pressure

Figure A1 shows results of the calibration of the best-fit parameter set in the model based on the flow height, velocity and pressure (Fig. A1a), only on maximum the flow height (Fig. A1b), only on the mean front velocity (Fig. A1c) and only on the maximum applied impact pressure (Fig. A1d).

5 ### Appendix B: Results of all the selected experiments and all the simulations

The results concerning the maximum flow height at position 2, the mean front velocity and the maximum impact pressure of all selected experiments and all the simulations we carried out are shown in Table B1.

*Competing interests.* No competing interests are present



**Table B1.** Values of the maximum flow height at position 2, the mean front velocity and the maximum applied pressure for all selected experiments and simulations. Bold text shows the maximum values of the height, the velocity and the pressure of all selected experiments

| Exp. No. / Simulation Parameters ($\phi_b$-$\epsilon_n$) | Max. flow height at pos. 2 (m) | Mean front velocity (m/s) | Max. pressure on large sensor (kPa) |
|---|---|---|---|
| Exp 9 | 0.29 | **10.2** | 65.9 |
| Exp 10 | **0.40** | 8.2 | 96.0 |
| Exp 11 | 0.38 | 9.0 | 94.6 |
| Exp 13 | 0.33 | 8.4 | 98.5 |
| Exp 14 | **0.40** | 9.1 | **138.0** |
| Exp 15 | 0.37 | 8.9 | 109.4 |
| Exp 16 | 0.37 | 6.4 | 69.2 |
| Mean value of experiments | 0.36 | 8.6 | 96.0 |
| Sim 20°- 0.30 | 0.32 | 9.7 | 169.3 |
| Sim 20°- 0.33 | 0.33 | 10.0 | 195.1 |
| Sim 20°- 0.36 | 0.34 | 10.1 | 133.1 |
| Sim 20°- 0.39 | 0.35 | 10.1 | 124.2 |
| Sim 20°- 0.42 | 0.35 | 10.2 | 99.9 |
| Sim 20°- 0.45 | 0.36 | 10.2 | 100.8 |
| Sim 25°- 0.30 | 0.36 | 8.7 | 131.2 |
| Sim 25°- 0.33 | 0.38 | 9.1 | 162.5 |
| Sim 25°- 0.36 | 0.39 | 9.4 | 89.9 |
| Sim 25°- 0.39 | 0.41 | 9.4 | 77.0 |
| Sim 25°- 0.42 | 0.42 | 9.6 | 70.6 |
| Sim 25°- 0.45 | 0.43 | 9.7 | 66.9 |
| Sim 30°- 0.30 | 0.38 | 8.2 | 97.1 |
| Sim 30°- 0.33 | 0.4 | 8.5 | 96.8 |
| Sim 30°- 0.36 | 0.41 | 9.0 | 83.9 |
| Sim 30°- 0.39 | 0.43 | 9.2 | 75.4 |
| Sim 30°- 0.42 | 0.44 | 9.2 | 67.6 |
| Sim 30°- 0.45 | 0.44 | 9.3 | 57.4 |
| Sim 35°- 0.30 | 0.41 | 7.4 | 95.1 |
| Sim 35°- 0.33 | 0.41 | 8.5 | 82.5 |
| Sim 35°- 0.36 | 0.42 | 8.8 | 76.4 |
| Sim 35°- 0.39 | 0.42 | 8.9 | 62.6 |
| Sim 35°- 0.42 | 0.43 | 9.2 | 58.0 |
| Sim 35°- 0.45 | 0.45 | 9.3 | 49.3 |
| Sim 40°- 0.30 | 0.45 | 7.0 | 98.7 |
| Sim 40°- 0.33 | 0.41 | 7.9 | 76.9 |
| Sim 40°- 0.36 | 0.41 | 8.5 | 70.2 |
| Sim 40°- 0.39 | 0.41 | 8.8 | 54.7 |
| Sim 40°- 0.42 | 0.43 | 8.9 | 56.0 |
| Sim 40°- 0.45 | 0.44 | 8.9 | 46.1 |



*Acknowledgements.* The research leading to these results has received funding from the Federal Office for the Environment (FOEN).





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
