# Peer review of "Elasto-plastic-adhesive DEM model for simulating hillslope debris flows: cross comparison with field experiments"

_Natural Hazards and Earth System Sciences, 2018_

## Short Comment (SC1) · 10 Dec 2018

In granular flow mechanics, grain size is a key quantity since it significantly affects flow mobility as its influence on basal pressure suggests. In this respect, I can confirm the effect of grain size on basal pressure observed by Albaba et al., because we too have obtained in both physical experiments and numerical simulations that the maximum (and mean) basal stresses increase as the mean grain size increases (the values of all other variables being constant).

This is important because it confirms that basal stresses are due to particle collisions and it is so irrespective of the presence of an interstitial mud, which is present in hillslope debris flows but absent from our dry granular flow that are meant to study pyroclastic flows and rock avalanches. In our laboratory experiments, we have measured the stresses at the base of dry granular flows that travel down a curved chute by means of a miniature load cell plate flush with the subsurface (Cagnoli and Romano, 2012b). These experimental results have then been generalized and shown to be scale invariant by Discrete Element Method simulations (Cagnoli and Piersanti, 2018).

Albaba et al. illustrate a good set of field experiments and DEM simulations. Therefore, I would like to underline that the collisions of single rock fragments take center stage as far as basal stresses are concerned. Here below a few comments.

1) I would add grain size data about the field slurries in line 16 on pag. 4.

2) Size and shape of the obstacles with the pressure sensors is a piece of information that must be provided. If these obstacles are not too tall with respect to the flow thickness, the recorded pressures can probably still be considered basal stresses. Is this the case? If different words are used in the text ("plate") and in Fig. 2 ("sensor"), it is not clear whether the pressure plates that have recorded the data are flush with the channel subsurface or they are those mounted on the obstacles protruding from the subsurface.

3) Filtering pressure data by replacing the original values with local averages (lines 29-30 pag. 4, line 1 pag. 5, lines 4-6 pag. 14) causes the loss of precious information about particle collisions. The same can be said about disregarding the data set from the smaller pressure plate (lines 14-16, pag. 10).

4) In our laboratory experiments finer grain size flow are faster than coarser ones (Cagnoli and Romano, 2012a). I therefore wonder whether the presence of an interstitial mud reduces the differential between the energy dissipation rates of flows with different grain size since you obtain virtually the same speed for them (lines 8-9, pag. 18). Unless the distance between position 2 and the release location is too small to see any difference.

[Figure]

5) Concerning the negative pressure values visible in Fig. 22 on pag. 23, are they artifacts due to pressure plate oscillations after collisions with the rock fragments?

Bruno Cagnoli, INGV

References

Cagnoli B. and Romano G.P. (2012a), Effects of flow volume and grain size on mobility of dry granular flows of angular rock fragments: A functional relationship of scaling parameters, J. Geophys. Res., 117, B02207, doi:10.1029/2011JB008926

Cagnoli B. and Romano G.P. (2012b), Granular pressure at the base of dry flows of angular rock fragments as a function of grain size and flow volume: A relationship from laboratory experiments, J. Geophys. Res., 117, B10202, doi:10.1029/2012JB009374

Cagnoli B. and Piersanti A. (2018), Stresses at the base of dry and dense flows of angular rock fragments in 3-D discrete element modeling: Scaling of basal stress fluctuations versus grain size, flow volume and channel width. J. Volcanol. Geotherm. Res., 349, 230-239.

---

## Referee Comment (RC1) · Lambert (Referee) · 20 Dec 2018

This paper addresses an interesting topic, and rather hot topic as studied by different research laboratories worldwide. This contribution is original and of real scientific value as it provides interesting insights into the DEM modeling of debris flow.

The two main interests in this article are, first, the use of a particular DEM contact law accounting for an adhesive force between particles and, second, the use of data from real-scale debris flow experiments for developing the model.

Even though this article is well written, well documented and constitutes an important

step forwards different comments rise after reading it. The detailed comments below globally invite the authors revising their article, while some may be considered as food for thought only.

To the referee's knowledge this is the very first publication proposing a DEM model of debris flow accounting for a tensile force between particles. Undeniably this is an interesting point. However, this topic is not really addressed in the article. The motivations for using such a contact law are not detailed nor argued, in particular with respect to other contact laws used for modeling debris flows. It is not explicitly stated what this law is supposed to allow accounting for. Besides, the consequences of using such a model on the granular flow behavior is not addressed in the results presentation neither than in the discussion. For sure, it has an influence on the flowing material velocity and height. It may also have a consequence on the interaction with the sensor (and thus on the impact force). Basically, the adhesive force may favor longer duration contacts between the particles and the wall (considering that contacting particles are pushed downward by other flowing particles). Considering the importance given to the type of contact law in the article title one could have expected more consideration to this crucial and innovative point. Experimental data from Bugnion et al. (2012) are used for developing the proposed model. More precisely, couples of parameters (basal friction angle, restitution coefficient) are calibrated against different flowing materials. In the end, the main conclusions drawn concern the ability of the modeling approach in satisfactorily fitting the experimental data, focusing on 3 measurements (flow height and velocity and pressure on the obstacle). So, contrary to what is suggested in the title, what is cross compared with field experiments is not the model itself, but the modeling approach. This comment is motivated by the fact that the calibration of the model parameters is conducted using these experimental data.

Concerning the area considered for measuring the pressure it seems the simulations do not perfectly meet the experimental conditions. For instance, the dimensions given in Bugnion et al. are the wedged dimensions and not the sensor dimensions. Differ-

ences thus seem to exist between the effective experimental measurements conditions and that in the simulations, with possible influence on results validity and discussion. The implication is that, for example, the sensor gives no value for thin flows and that the height of flow concerned by the measurement is not 295 mm, but much less. As dealing with a debris flow, a velocity gradient may be observed from the flow bottom to top, implying a variable impact force on the obstacle. The position and dimensions of the measuring surface should thus be identical between the experiments and the simulations. This point should be clarified.

Table 1 shows a rather large panel of experiments, varying three parameters related to the flowing material (wet density, water mass fraction and fine mass content). Due to these differences, very variable values where measured in relation to the flowing material and impact pressure. But in such a context, it doesn't seem relevant to refer to mean values when comparing the results of all the tests, as done section 3. The initial conditions are extremely variable and consequently the velocities, flow thickness and pressure differ significantly from one test to the other: a comparison based on an average value seems to be of very limited interest and relevance. In addition, and due to these differences in initial material characteristics, tests 9, 14 and 16 seem to pose a problem to the authors, either when comparing the maximum pressure of the different tests or when comparing the experimental data with DEM results. A basic way to compare these results, is to compute the hydrodynamic pressure which is proportional to (unit mass*$v^2$) and to compare this pressure with that measured (as done by Bugnion et al.). When plotting this term versus Pmax, it appears that most of the test results are aligned, with a ratio of 0.76+/-0.1 between the two parameters. The exceptions are tests 9 (ratio of 0.35) and, to a lesser extent, test 11 (ratio of 0.57). This suggests that the pressure measured for test 14 is in line with other experiments, when considering the unit mass and the velocity, and may not be justified by the presence of large boulders in the flow. The difficulty for the DEM model to well reproduce results of tests 9, 14 and 16 certainly finds an explanation in Figure 16. This figure shows that these tests are far from the domain where the modeling approach gives good results. In other words, Figure 16 reveals the domain where the proposed DEM modeling approach is valid, in terms of restitution coefficient and basal friction angle and after having calibrated the model parameters. Cases 9, 14 and 16 are out of this validity domain.

Similarly as for the experiments, a 50ms filtering interval was used to smooth the pressure curves. This temporal window aims at smoothing sensor plate vibrations resulting from the impact of solid grains (Bugnion et al.). It was observed that after an impact by a single solid ball the sensor plates vibrates for up to 30 ms. This period of time is much longer than the impact duration : it takes a few milliseconds for having a momentum transfer from the solid ball to the plate. In the case of debris flows, these peaks come in addition to pressure transmitted by the matrix surrounding large particles (this matrix consists in a mixture of water and fine grains). Bugnion et al. stated that this technique was efficient in smoothing peaks due to solid grains contained in the debris flows, without altering the information. Nevertheless, it doesn't seem really justified to consider the same filtering interval for treating the DEM results because there is no plate vibration but just short duration peaks (2ms) related to the plate-particle contacts. Such peaks do not justify having a 50ms filtering interval. This comment is also motivated by the point addressed below.

The DEM model considers an assembly of large grains, with mean diameters ranging from 75 to 150 mm. For the main case, the diameter ranges from 50 to 100mm, which represent only a fraction of the grain-size distribution of the debris flows considered by Bugnion et al (less than 30% in mass). Considering the size of the sensor it appears that the maximum number of d50-particles in contact with the plate goes from about 12 for d50 = 75 mm down to less than 4 for d50 = 150 mm. Such a small number of contacts has a strong influence on the pressure deduced from the DEM simulations. In the absence of matrix, the force exerted by the flow on the sensor is then the sum of a small number of short duration contacts. As the interaction with the sensor is a key issue in this work, a more in-depth investigation of the particle-plate interaction could

have been conducted, for instance addressing items such as pressure representativeness (variability in results repeating the test varying the initial particles packing), peak force amplitude, impact duration, number of contacts, . . .. This is particularly critical for d50 = 150 mm, and results presented in Fig. 17 may be explained by the fact that the variability in force exerted on the plate and resulting from the small number of contact points has not been accounted for. The influence of the small number of contact points should also be checked for cases down to d50=75mm.

The model is calibrated focusing on 2 parameters: the basal friction angle and the ratio of k1 to k2. This later is referred to as restitution coefficient but this term seems improper. It doesn't correspond to the classical definition of the restitution coefficient (ratio of velocities between after and before contact) and thus may introduce ambiguity. This comment seems to be supported by the results plotted in Fig. 18, revealing the very limited influence of this coefficient on the flow velocity and height. For what concerns the basal friction angle, this parameter seems artificial. Figure 16 shows that for a same slope, the basal friction angle varies from 25 to 40° depending on the flowing material. This range may hardly be justified by changes in the slope characteristics from one test to the other, neither than the 40° value be justified. It seems on the contrary that this parameter is a way to account for the flowing material rheology. The debris flows being modeled as a collection of particles of large diameter with respect to the flow thickness, a good agreement of the DEM model with the experiments in terms of flow height and depth-averaged velocity requires adjusting the basal friction angle. This parameter is thus not intrinsic to the channel characteristics, but also accounts for the rheology of the debris flows. This would deserve specific comments and probably discussion.

On a formal point of view the paper organization is sometimes confusing. The calibrated parameters are presented late, after plotting and discussing the results. Technical details concerning the experiments are presented in the discussion, while these pieces of information were discussed by Bugnion et al. and the figure was directly

copied from their article. As such, this should be introduced together with the experiments, in section 2. Last, in section 3.3 dedicated to the parametric study, there is a mix between DEM model parameters and physical parameters describing the flowing material and conditions. Considering the former or the later does not fall within the same scope.

Last, curves plotted Figures 13 to 15 do not correspond to the maximum values plotted in Figure 11.

---

## Referee Comment (RC2) · Alessandro Leonardi (Referee) · 19 Mar 2019

The paper does an excellent job in highlighting the importance and urgency of the research presented. The review of the state of the art is complete and clear. The language is good, and while some errors and typos are present in the manuscript, they do not reduce the readability of the paper. The comparison with full-scale experimental recordings is a very strong aspect of the paper. Comparison with field measurements is particularly difficult for mass flows, and makes the results much more credible.

The choice of testing a DEM model with cohesion is of great interest, because it shows a different perspective on a long-standing problem in the field. In fact, it is widely known

that standard DEM models (i.e. with dry, adhesionless particles) have strong limitation in the simulation of water-rich debris flow.

I found the article pleasurable to read. However, many points need further clarifications, and possibly a few more simulations would much strengthen the impact of the study. I list here the major issues:

- The adoption of a non-standard contact model, while being the most interesting aspect of the work, is not followed by an appropriate analysis of its capabilities. The varied parameters only include friction and restitution, which are parameters that would intuitively be chosen for adhesionless granular assemblies. I would have suggested to also test the influence of parameters such as the "minimum force", which based on Eq. 1 and discussion, seem to control the adhesive part of the model.

- Many problematic aspects of the simulations, such as the absence of a deposit, or the creation of a dilute front, could effectively be reduced by exploiting the adhesive bonds between the particles. However, this is not addressed in depth in the paper.

- With respect to the lack of deposit, it is not clear why simulations with a friction angle larger than the flume slope ($40°$ vs $30°$) do not produce a deposit. As a matter of fact, I would have expected to see little mass mobilization in this case. Adhesion would have further prevented the mass from mobilizing. In my opinion, this aspect should be clearly addressed in the paper.

- Overall, the choice of the simulation parameters could be motivated more. The values of mean particle diameter, particle density, inter-particle friction, and Young modulus are given without a convincing explanation behind their choice.

- The analysis of Fig. 8-15 and 17-20 is mostly descriptive, and does not add much to the figures themselves. The authors possess a lot of information that they do not use for the interpretation of the results. For example, in section 3.3.2, the authors guess that the excessive dispersion is due to the decrease of plastic deformation. However, the

authors do have the information to check if this is indeed the case. Here again, it would have been very interesting to see whether the adhesive bonds could have countered this unwanted effect. Another example is in section 3.3.3, where the authors suppose that the augmented flow shearing due to an increase of phi_b generates thicker flows. Once more, the authors could check whether this is the case, rather than leave the interpretation open. Here adhesion is one more not mentioned. If, instead of basal shear resistance, the adhesive bond would have been boosted, would the results have changed in a less intuitive fashion?

- In the comparison between measurement of basal pressure and impact load on the sensor, clearly the applied filter has a great influence on the results. The authors do a good job at describing the problems associated with this. However, the particle size chosen for the simulation does not correspond to the one used in the experiments. Therefore, applying the same smoothening window as in the experiments does not seem like an intuitive choice. Since pressure is one of the calibration parameters, this might lead to erroneous results, and maybe partially explaining the difficulties in obtaining a better calibration for both velocity and flow height.

- Finally, I think that the paper would benefit form a reorganization of section 3. The results of the sensitivity analysis (3.3) should be presented before the comparison with experimental values.

Overall, the paper is set out to offer a new and very interesting approach for the simulation of debris flow. However, the capabilities of the most innovative part of the proposed model are not really explored. The paper offers an analysis that reiterates some common findings in the literature. In fact, difficulties in reproducing flow mobility and in obtaining a correct estimation of impact forces are very common. It would have been very interesting to see if adhesion helps in addressing these long-standing problems. However, the paper does not offer much insight in this respect.

Minors: Page 2, last line: "flowing velocity, flowing height" maybe better "flow velocity,

flow height" Page 3, line 25: "and and" Page 4, line 21: "channelized channels" sounds weird Page 8, line 13 "chut's bottom" Fig.5 The picture bottoms are cropped before the end of the chute. Fig. 8: when printed in grayscale, the lines become very similar. Fig. 22 Row -> Raw. Also: why negative values? Page 13, line 6: well -> good

---

## Author Comment (AC1) · 14 Jun 2019

The authors would like to sincerely thank Dr. Stéphane Lambert for his comments and suggestions on the submitted research paper. Each comment and its corresponding reply are addressed in the following text.

*This paper addresses an interesting topic, and rather hot topic as studied by different research laboratories worldwide. This contribution is original and of real scientific value as it provides interesting insights into the DEM modeling of debris flow.*

The authors thank the referee for confirming the importance and relevance of the topic.

*To the referee's knowledge this is the very first publication proposing a DEM model of debris flow accounting for a tensile force between particles. Undeniably this is an interesting point. However, this topic is not really addressed in the article. The motivations for using such a contact law are not detailed nor argued, in particular with respect to other contact laws used for modeling debris flows. It is not explicitly stated what this law is supposed to allow accounting for. Besides, the consequences of using such a model on the granular flow behavior is not addressed in the results presentation neither than in the discussion. For sure, it has an influence on the flowing material velocity and height. It may also have a consequence on the interaction with the sensor (and thus on the impact force). Basically, the adhesive force may favor longer duration contacts between the particles and the wall (considering that contacting particles are pushed downward by other flowing particles). Considering the importance given to the type of contact law in the article title one could have expected more consideration to this crucial and innovative point.*

The introductory part of the article included references to previous DEM contact laws that were developed to simulate granular flows down inclined planes. Some of these models considered dry granular flows while others included the effect of water by coupling the classical visco-elastic DEM model with a fluid-solver such as LBM or CFD. However, the latter was found to be computationally very expensive and its application for practical granular flows is limited. In order to further clarify our motivation for using this adhesive elasto-plastic model, a paragraph will be added at the end of the introduction section which states:
"This paper presents a new computationally-efficient DEM model that would partially account for the presence of the fluid composed of water and find material, based on the work of (Luding, 2008). This is achieved through the adhesive aspect of the contact law which would indirectly take the presence of such fluid into account, as this fluid would increase the cohesion of the flowing mass. The advantage of this new approach is that it accounts for the interaction between solid grains of the flowing mass as well as the effect of fluid between them, all in the same modeling frame (DEM). As a result, modeling 3D real scale experiments or back-calculating historical events of granular flows would be computationally possible.

Furthermore, in the revised version, a clear difference will be highlighted between the particle-particle interaction and the particle-wall interaction, with the "wall" object being either the sensor or the channel base. The authors will clearly state that the particle-particle interaction is governed by the new proposed contact law of Luding (2008). On the other side, the particle-wall interaction is governed by the classical visco-elastic contact law (Schwager and Pöschel, 2007) for which previous studies of Albaba et al. (2015) has been used as reference for calibrating its parameter values concerning the impact between a flowing mass and a rigid wall.

In addition, in the revised version, the effect of the possible formation of adhesive bonds and their effect on the results concerning height, velocity and pressure will be discussed by selecting a virtual sampling box at a fixed distance away from the sensor where such bonds will be characterized and compared for different model parameter combinations.

*Experimental data from Bugnion et al. (2012) are used for developing the proposed model. More precisely, couples of parameters (basal friction angle, restitution coefficient) are calibrated against different flowing materials. In the end, the main conclusions drawn concern the ability of the modeling approach in satisfactorily fitting the experimental data, focusing on 3 measurements (flow height and velocity and pressure on the obstacle). So, contrary to what is suggested in the title, what is cross compared with field experiments is not the model itself, but the modeling approach. This comment is motivated by the fact that the calibration of the model parameters is conducted using these experimental data.*

In the revised version, the title will be changed to in order to better reflect the content of the article.

*Concerning the area considered for measuring the pressure it seems the simulations do not perfectly meet the experimental conditions. For instance, the dimensions given in Bugnion et al. are the wedged dimensions and not the sensor dimensions. Differences thus seem to exist between the effective experimental measurements conditions and that in the simulations, with possible influence on results validity and discussion. The implication is that, for example, the sensor gives no value for thin flows and that the height of flow concerned by the measurement is not 295 mm, but much less. As dealing with a debris flow, a velocity gradient may be observed from the flow bottom to top, implying a variable impact force on the obstacle. The position and dimensions of the measuring surface should thus be identical between the experiments and the simulations. This point should be clarified.*

The authors have obtained internal reports of the tests where the precise measurements of the sensors are written. The large sensor has a size of 20x20 cm while the small one has a size of 12x12 cm. These new details concerning the sensors sizes will be taken into account when carrying simulations for the revised version. The sizes of plates representing the sensors in DEM will be modified accordingly.
On the other hand, this raised point by the referee explains the phenomenon seen when comparing the evolution of pressure with time for the experiments and the model. The model is found to register pressure values earlier than the experiment. This now can be explained further by the fact that the experiments do not register pressure values for thin flows or for flows with heights lower than the height of the bottom edge of the sensor. On the other hand, the DEM model starts registering values for all types of interaction even for single particles at the beginning. This point will be further discussed in the revised manuscript of the article.

*Table 1 shows a rather large panel of experiments, varying three parameters related to the flowing material (wet density, water mass fraction and fine mass content). Due to these differences, very variable values where measured in relation to the flowing material and impact pressure. But in such a context, it doesn't seem relevant to refer to mean values when comparing the results of all the tests, as done section 3. The initial conditions are extremely variable and consequently the velocities, flow thickness and pressure differ significantly from one test to the other: a comparison based on an average value seems to be of very limited interest and relevance.*

The main purpose of presenting the results in mean values in Figure 8 was to show how do flow velocity, flow height and pressure fluctuate around the mean for different experiments and different initial conditions. The mean value of different experiments was not used for any comparison with numerical data, since that one of the objectives of the article is to relate the model parameters with the initial conditions of the flowing mass.
However, to avoid confusion, Figure 8 will be deleted in the revised version.

*In addition, and due to these differences in initial material characteristics, tests 9, 14 and 16 seem to pose a problem to the authors, either when comparing the maximum pressure of the different tests or when comparing the experimental data with DEM results. A basic way to compare these results, is to compute the hydrodynamic pressure which is proportional to (unit mass \* v²) and to compare this pressure with that measured (as done by Bugnion et al.). When plotting this term versus Pmax, it appears that most of the test results are aligned, with a ratio of 0.76+/-0.1 between the two parameters. The exceptions are tests 9 (ratio of 0.35) and, to a lesser extent, test 11 (ratio of 0.57). This suggests that the pressure measured for test 14 is in line with other experiments, when considering the unit mass and the velocity, and may not be justified by the presence of large boulders in the flow. The difficulty for the DEM model to well reproduce results of tests 9, 14 and 16 certainly finds an explanation in Figure 16. This figure shows that these tests are far from the domain where the modeling approach gives good results. In other words, Figure 16 reveals the domain where the proposed DEM modeling approach is valid, in terms of restitution coefficient and basal friction angle and after having calibrated the model parameters. Cases 9, 14 and 16 are out of this validity domain.*

Tests 9, 14 and 16 are of special interest for the authors since they have considerably different values of pressures in comparison with other tests. However, those values of pressures were well reproduced by the corresponding DEM simulations as seen in Figure 11.

Bugnion et al. (2012) carried detailed analysis of the pressure measured by both sensors and related it to the hydrodynamic pressure measured as the product of mass and squared front velocity. This however is not in the core interest of the comparison for the authors since such calculated pressure is only proportional to the measured ones. The calibration process has an objective of reproducing real pressure values by DEM simulation. However, in the revised version, a new column will be added to Table 1 representing the hydrodynamic pressure of each test as presented by Bugnion et al. (2012). In addition, a paragraph will be added to stress the fact that the difference in pressure of tests 9, 14 and 16 is best explained by a change in the hydrodynamic pressure. However, the explanation of the pressure difference between tests 14 and 9 is also possible by assuming the impact of large particles on the sensor in case of test 14 when compared with test 9. This is because the gravel percentage of the mixture in test 14 is 3 times higher than that of test 9 (see Table 3 in Bugnion et al. 2012).

Concerning Figure 16, it is meant to illustrate the sets of model parameters that are used to reproduce the different experimental tests, depending on the chosen calibration process. This figure shows by no means the domain where the proposed DEM model is valid. In contrast, it shows for each experimental test the right values of φ and ε to be used to reproduce that test. A reader can be misled by the red dashed-line drawn in Figure 16 which purpose was to limit a zone where several experimental tests are reproduced with very similar set of parameters of DEM model. Such confusion will be avoided when preparing the revised manuscript by eliminating that line and by better introducing the figure in the text.

*Similarly as for the experiments, a 50ms filtering interval was used to smooth the pressure curves. This temporal window aims at smoothing sensor plate vibrations resulting from the impact of solid grains (Bugnion et al.). It was observed that after an impact by a single solid ball the sensor plates vibrates for up to 30 ms. This period of time is much longer than the impact duration: it takes a few milliseconds for having a momentum transfer from the solid ball to the plate. In the case of debris flows, these peaks come in addition to pressure transmitted by the matrix surrounding large particles (this matrix consists in a mixture of water and fine grains). Bugnion et al. stated that this technique was efficient in smoothing peaks due to solid grains contained in the debris flows, without altering the information. Nevertheless, it doesn't seem really justified to consider the same filtering interval for treating the DEM results because there is no plate vibration but just short duration peaks (2ms) related to the plate-*

*particle contacts. Such peaks do not justify having a 50ms filtering interval. This comment is also motivated by the point addressed below.*

*The DEM model considers an assembly of large grains, with mean diameters ranging from 75 to 150 mm. For the main case, the diameter ranges from 50 to 100mm, which represent only a fraction of the grain-size distribution of the debris flows considered by Bugnion et al (less than 30% in mass). Considering the size of the sensor it appears that the maximum number of d50-particles in contact with the plate goes from about 12 for d50 = 75 mm down to less than 4 for d50 = 150 mm. Such a small number of contacts has a strong influence on the pressure deduced from the DEM simulations. In the absence of matrix, the force exerted by the flow on the sensor is then the sum of a small number of short duration contacts. As the interaction with the sensor is a key issue in this work, a more in-depth investigation of the particle-plate interaction could have been conducted, for instance addressing items such as pressure representativeness (variability in results repeating the test varying the initial particles packing), peak force amplitude, impact duration, number of contacts, …This is particularly critical for d50 = 150 mm, and results presented in Fig. 17 may be explained by the fact that the variability in force exerted on the plate and resulting from the small number of contact points has not been accounted for. The influence of the small number of contact points should also be checked for cases down to d50=75mm.*

The authors thank the referee for raising this important point concerning the filtering of DEM signal which has a direct effect on the way the maximum impact pressure is calculated and also relates to the calibration process which is partially based on the comparison between maximum pressures of DEM and Exp, in addition to the flowing height and velocity.

The strong oscillations of DEM signals are usually linked to many factors including the number of particles, the area that is being impacted, the frequency of recording data, the mean particle diameter and the number of contacts.

One difficulty in the current study however is the fact that the experiment represents a full-scale hill slope debris flow with a volume of 50 m$^3$. Such a large volume requires running simulation with particle sizes that are relatively large ($d_{50}$ = 75 mm) in comparison with the range of sizes, in order to keep the total number of particles within feasible range as to the computation capabilities of the super computers (the average total number of particles is around 160,000). In addition, the sensor size is indeed small (200x200 mm) in comparison to the mean particle size considered for the simulations, which leads to having few contacts per impacting step and thus a discrete fluctuating signal in DEM.

Furthermore, the possible variation of the particles' initial spatial distribution in the released material might also have a small effect on the force signal, as reported in some DEM studies (e.g. Albaba et al 2015).

Because of all aforementioned reasons, there is a need to define a filtering interval based solely on an investigation of the DEM signal and independent of the experiment's filtering interval.

In the revised version, the authors propose an analysis of the DEM signal based on two points:
   I.    The repeatability of the same tests to account for initial spatial variation.
   II.   The signal of different simulations with different parameter values.

First, the same DEM simulation would be run 10 times with different initial spatial distribution and then the maximum pressure will be plotted against different filtering intervals (0.025 s up to 0.5 s). In addition, the relative error defined as the normalized difference between two successive values of maximum impact pressure would be plotted. An optimum filtering interval would be defined as that with a relative error of 5% or lower. The same would then be carried out for the different DEM simulations with different parameters of φ and ε.

The filtering interval to be used for filtering the DEM and deducing the maximum impact pressure would be the optimal one while considering the two points above. An example of the proposed analysis is presented below in Fig R1 for a simulation with $\varphi = 30°$ and $\varepsilon = 0.3$

[Figure]

*Figure R1 The maximum pressure and relative error for different filtering intervals for DEM simulation with $\varphi = 30°$ and $k_1/k_2 = 0.3$.*

Furthermore, an in-depth analysis will be carried out concerning the effect of the particle size (75, 100, 125 and 150 mm) on the impact pressure signal by relating mean particle diameter, evolution of number of contacts and the duration of the contact.

*The model is calibrated focusing on 2 parameters: the basal friction angle and the ratio of k1 to k2. This later is referred to as restitution coefficient but this term seems improper. It doesn't correspond to the classical definition of the restitution coefficient (ratio of velocities between after and before contact) and thus may introduce ambiguity. This comment seems to be supported by the results plotted in Fig. 18, revealing the very limited influence of this coefficient on the flow velocity and height. For what concerns the basal friction angle, this parameter seems artificial. Figure 16 shows that for a same slope, the basal friction angle varies from 25 to 40_ depending on the flowing material. This range may hardly be justified by changes in the slope characteristics from one test to the other, neither than the 40_ value be justified. It seems on the contrary that this parameter is a way to account for the flowing material rheology. The debris flows being modeled as a collection of particles of large diameter with respect to the flow thickness, a good agreement of the DEM model with the experiments in terms of flow height and depth-averaged velocity requires adjusting the basal friction angle. This parameter is thus not intrinsic to the channel characteristics, but also accounts for the rheology of the debris flows. This would deserve specific comments and probably discussion.*

The use of the term restitution coefficient confirms the choice of terms of Luding (2008). It is however true that it is also usually used to refer to the ratio of velocities before and after the impact for classical visco-elastic contact laws, such as that of Schwager and Pöschel (2007). It will be labelled as $k_1/k_2$ in the new version of paper to avoid ambiguity. Concerning the friction angle, it would be recalled "microscopic" friction angle since it defines the friction angle at the microscopic scale between each single particle and the base of the channel.

The effect of the ratio of $k_1/k_2$ on the flow height and velocity was found to be limited because the variation of this ratio only concerns the interaction between the particles themselves. The interaction between flowing particles and the base of the channel is governed by a visco-elastic contact law where the value of $\varepsilon_n$ was fixed based on previous studies of Albaba et al (2015). In that study, it was shown that the flow height is affected by the value of restitution coefficient while the flow velocity is governed by the value of the microscopic friction angle.

All these details will be presented in detail in the new version and critically analyzed.

On a formal point of view the paper organization is sometimes confusing. The calibrated parameters are presented late, after plotting and discussing the results. Technical details concerning the experiments are presented in the discussion, while these pieces of information were discussed by Bugnion et al. and the figure was directly copied from their article. As such, this should be introduced together with the experiments, in section 2. Last, in section 3.3 dedicated to the parametric study, there is a mix between DEM model parameters and physical parameters describing the flowing material and conditions. Considering the former or the later does not fall within the same scope.

As stated in the previous authors' replies, the new version of the paper will be prepared in compliance with the proposition of the referee. After the introduction, a detailed sensitivity analysis of the model's parameters will be presented in addition to the analysis of the filtering interval. After optimizing the choice of filtering interval of DEM, a comparison between the model and experiment will be detailed. The discussion will then be based on the analysis of pressure signal in addition to further analysis of the comparison between DEM and Exp data. All pieces of information concerning the experiment will be introduced when introducing the experimental data.

Last, curves plotted Figures 13 to 15 do not correspond to the maximum values plotted in Figure 11.

Values concerning the maximum impact forces (after applying the 0.05 seconds filtering interval) of the experiments of Bugnion et al. (2012) are given in Figure 11 and table 1 in the paper. The temporal evolution of pressure of the different experimental tests (Fig 12 to Fig 15) were reproduced from non-published internal reports of the experiments. This will be cleared in the new version of the paper.

---

## Author Comment (AC2) · 14 Jun 2019

The authors would like to sincerely thank Dr. Alessandro Leonardi for his comments and suggestions on the submitted research paper. Each comment and its corresponding reply are addressed in the following text.

*The paper does an excellent job in highlighting the importance and urgency of the research presented. The review of the state of the art is complete and clear. The language is good, and while some errors and typos are present in the manuscript, they do not reduce the readability of the paper. The comparison with full-scale experimental recordings is a very strong aspect of the paper. Comparison with field measurements is particularly difficult for mass flows, and makes the results much more credible. The choice of testing a DEM model with cohesion is of great interest, because it shows a different perspective on a long-standing problem in the field. In fact, it is widely known that standard DEM models (i.e. with dry, adhesionless particles) have strong limitation in the simulation of water-rich debris flow.*

The authors thank the referee for confirming the importance and relevance of the topic.

*I found the article pleasurable to read. However, many points need further clarifications, and possibly a few more simulations would much strengthen the impact of the study. I list here the major issues:*
*-        The adoption of a non-standard contact model, while being the most interesting aspect of the work, is not followed by an appropriate analysis of its capabilities. The varied parameters only include friction and restitution, which are parameters that would intuitively be chosen for adhesionless granular assemblies. I would have suggested to also test the influence of parameters such as the "minimum force", which based on Eq. 1 and discussion, seem to control the adhesive part of the model.*

In the sensitivity study of the model, different parameters have been investigated such as the microscopic friction angle and the restitution coefficient. However, the restitution coefficient is meant here as the ratio of $k_1/k_2$ (Luding, 2008). Although it shares the same name, it is not the same as for cohesion-less models which defines the ratio of velocities before and after an impact of two objects (Schwager and Pöschel, 2007). Thus, to avoid ambiguity, the term $k_1/k_2$ would be used in the revised version instead of restitution coefficient. A more in-depth analysis of the model parameters will also be carried out including testing the effect of $k_c$ which defines the minimum force of Eq. 1.

*-        Many problematic aspects of the simulations, such as the absence of a deposit, or the creation of a dilute front, could effectively be reduced by exploiting the adhesive bonds between the particles. However, this is not addressed in depth in the paper.*

The presence of a dilute front would change depending on the value of $k_1/k_2$ which partially controls the activation of adhesive bonds. If the adhesive bonds are not activated, particles would detach from the flow and a dilute front would appear. The lack of deposition is addressed in the next comment.

*-        With respect to the lack of deposit, it is not clear why simulations with a friction*

*angle larger than the flume slope (40° vs 30°) do not produce a deposit. As a matter of fact, I would have expected to see little mass mobilization in this case. Adhesion would have further prevented the mass from mobilizing. In my opinion, this aspect should be clearly addressed in the paper.*

In the DEM model, the friction angle value is meant as the value of friction angle between a sliding particle and the channel base. Since this value is calculated at a microscopic level, it does not correspond to the macroscopic value of friction angle which would have stopped the mass from sliding if it had exceeded the channel's inclination angle (30°). In order to avoid ambiguity, the friction angle in the paper would be referred to as the microscopic friction angle and would be stressed that it is the one calculated at the micro-scale between sliding particles and the channel base. Furthermore, in the revised version, a clear difference will be highlighted between the particle-particle interaction and the particle-wall interaction, with the wall being either the sensor or the channel base. The authors will clearly state that the particle-particle interaction is governed by the new proposed contact law of Luding (2008). On the other side, the particle-wall interaction is governed by the classical visco-elastic contact law (Schwager and Pöschel, 2007) for which previous studies of Albaba et al. (2015) have been used as reference for calibrating its parameter values concerning the impact between a flowing mass and a rigid wall.

*- Overall, the choice of the simulation parameters could be motivated more. The values of mean particle diameter, particle density, inter-particle friction, and Young modulus are given without a convincing explanation behind their choice.*

The choice of the model parameters was based on previous simulations of Albaba et al. (2015). It would be clearly stated in the revised version.

*- The analysis of Fig. 8-15 and 17-20 is mostly descriptive, and does not add much to the figures themselves. The authors possess a lot of information that they do not use for the interpretation of the results. For example, in section 3.3.2, the authors guess that the excessive dispersion is due to the decrease of plastic deformation. However, the authors do have the information to check if this is indeed the case. Here again, it would have been very interesting to see whether the adhesive bonds could have countered this unwanted effect. Another example is in section 3.3.3, where the authors suppose that the augmented flow shearing due to an increase of phi_b generates thicker flows. Once more, the authors could check whether this is the case, rather than leave the interpretation open. Here adhesion is one more not mentioned. If, instead of basal shear resistance, the adhesive bond would have been boosted, would the results have changed in a less intuitive fashion?*

In addition to the investigated parameters, a detailed investigation will be carried out regarding the impact pressure signal and its relation to the filtering interval (more information in the next comment). Furthermore, an in-depth analysis will be carried out concerning the effect of the particle size (75, 100, 125 and 150 mm) by relating mean particle diameter, evolution of number of contacts and the duration of the contact to the observed flow behavior. Moreover, the evolution of the cohesive bonds with time

for different contacts within the flow will be analyzed and the possible relation to the model's parameters will be discussed.

*- In the comparison between measurement of basal pressure and impact load on the sensor, clearly the applied filter has a great influence on the results. The authors do a good job at describing the problems associated with this. However, the particle size chosen for the simulation does not correspond to the one used in the experiments. Therefore, applying the same smoothening window as in the experiments does not seem like an intuitive choice. Since pressure is one of the calibration parameters, this might lead to erroneous results, and maybe partially explaining the difficulties in obtaining a better calibration for both velocity and flow height.*

The authors thank the referee for raising this important point concerning the filtering of DEM signal which has a direct effect on the way the maximum impact pressure is calculated and also relates to the calibration process which is partially based on the comparison between maximum pressures of DEM and Exp, in addition to the flowing height and velocity.

The strong oscillations of DEM signals are usually linked to many factors including the number of particles, the area that is being impacted, the frequency of recording data, the mean particle diameter and the number of contacts.

One difficulty in the current study however is the fact that the experiment represents a full-scale hill slope debris flow with a volume of 50 m$^3$. Such a large volume requires running simulation with particle sizes that are relatively large ($d_{50}$ = 75 mm) in comparison with the range of sizes, in order to keep the total number of particles within feasible range as to the computation capabilities of the super computers (the average total number of particles is around 160,000). In addition, the sensor size is indeed small (200x200 mm) in comparison to the mean particle size considered for the simulations, which leads to having few contacts per impacting step and thus a discrete fluctuating signal in DEM.

Furthermore, the possible variation of the particles' initial spatial distribution in the released material might also have a small effect on the force signal, as reported in some DEM studies (e.g. Albaba et al 2015).

Because of all aforementioned reasons, there is a need to define a filtering interval based solely on an investigation of the DEM signal and independent of the experiment's filtering interval.

In the revised version, the authors propose an analysis of the DEM signal based on two points:
 I.   The repeatability of the same tests to account for initial spatial variation.
 II.  The signal of different simulations with different parameter values.

First, the same DEM simulation would be run 10 times with different initial spatial distribution and then the maximum pressure will be plotted against different filtering intervals (0.025 s up to 0.5 s). In addition, the relative error defined as the normalized difference between two successive values of maximum impact pressure would be plotted. An optimum filtering interval would be defined as that with a relative error of 5% or lower. The same would then be carried out for the different DEM simulations with different parameters of $\varphi$ and $\varepsilon$.

The filtering interval to be used for filtering the DEM and deducing the maximum impact pressure would be the optimal one while considering the two points above. An example of the proposed analysis is presented below in Fig R1 for a simulation with φ = 30° and ε = 0.3

[Figure]

*Figure R1 The maximum pressure and relative error for different filtering intervals for DEM simulation with φ = 30° and $k_1/k_2$ = 0.3.*

-        *Finally, I think that the paper would benefit form a reorganization of section 3. The results of the sensitivity analysis (3.3) should be presented before the comparison with experimental values.*

The revised version will be prepared in compliance with the proposition of the referee. After the introduction, a detailed sensitivity analysis of the model's parameters will be presented in addition to the analysis of the filtering interval. After optimizing the choice of filtering interval of DEM, a comparison between the model and experiment will be detailed. The discussion will then be based on the analysis of pressure signal in addition to further analysis of the comparison between DEM and Exp data. All pieces of information concerning the experiment will be introduced when introducing the experimental data.

*Minors: Page 2, last line: "flowing velocity, flowing height" maybe better "flow velocity, flow height" Page 3, line 25: "and and" Page 4, line 21: "channelized channels" sounds weird Page 8, line 13 "chut's bottom" Fig.5 The picture bottoms are cropped before the end of the chute. Fig. 8: when printed in grayscale, the lines become very similar.*

*Fig. 22 Row -> Raw. Also: why negative values? Page 13, line 6: well -> good*

All these minor points will be corrected in the revised version

---

## Author Comment (AC3) · 14 Jun 2019

The authors would like to sincerely thank Dr. Bruno Cagnoli for his comments on the submitted research paper.

The comparison between the numerical DEM model and the experiments concern the flowing velocity, flow height and the impact pressure that is exerted by the flow on the vertically mounted plate/sensor. The basal friction however has not been investigated in this paper due to the lack of reliable experimental details.

The effect of grain size is indeed an important factor to be considered, especially for

[Figure]

DEM simulations which are based on the discrete interaction between objects. As a result, different particle size values have been tested in the sensitivity study of the model and their respective effect on the flow behaviour and impact pressure have been discussed.

1) I would add grain size data about the field slurries in line 16 on pag. 4.

The grain size data is added in the revised version

2) Size and shape of the obstacles with the pressure sensors is a piece of information that must be provided. If these obstacles are not too tall with respect to the flow thickness, the recorded pressures can probably still be considered basal stresses. Is this the case? If different words are used in the text ("plate") and in Fig. 2 ("sensor"), it is not clear whether the pressure plates that have recorded the data are flush with the channel subsurface or they are those mounted on the obstacles protruding from the subsurface.

The new version of the paper will clearly state that the sensors are installed in the vertical direction to the channel base and flow direction. As such, the pressure recorded during the experiments and simulations are impact pressure in normal to the sensors. The word plate would not be used in the revised version

3) Filtering pressure data by replacing the original values with local averages (lines 29-30 pag. 4, line 1 pag. 5, lines 4-6 pag. 14) causes the loss of precious information about particle collisions. The same can be said about disregarding the data set from the smaller pressure plate (lines 14-16, pag. 10).

The authors are thankful for raising this important point concerning the filtering of DEM signal which has a direct effect on the way the maximum impact pressure is calculated and also relates to the calibration process which is partially based on the comparison between maximum pressures of DEM and Exp, in addition to the flowing height and velocity. The strong oscillations of DEM signals are usually linked to many factors

including the number of particles, the area that is being impacted, the frequency of recording data, the mean particle diameter and the number of contacts. One difficulty in the current study however is the fact that the experiment represents a full-scale hill slope debris flow with a volume of 50 m3. Such a large volume requires running simulation with particle sizes that are relatively large (d50 = 75 mm) in comparison with the range of sizes, in order to keep the total number of particles within feasible range as to the computation capabilities of the super computers (the average total number of particles is around 160,000). In addition, the sensor size is indeed small (200x200 mm) in comparison to the mean particle size considered for the simulations, which leads to having few contacts per impacting step and thus a discrete fluctuating signal in DEM. Furthermore, the possible variation of the particles' initial spatial distribution in the released material might also have a small effect on the force signal, as reported in some DEM studies (e.g. Albaba et al 2015). Because of all aforementioned reasons, there is a need to define a filtering interval based solely on an investigation of the DEM signal and independent of the experiment's filtering interval. In the revised version, the authors propose an analysis of the DEM signal based on two points: I. The repeatability of the same tests to account for initial spatial variation. II. The signal of different simulations with different parameter values.

First, the same DEM simulation would be run 10 times with different initial spatial distribution and then the maximum pressure will be plotted against different filtering intervals (0.025 s up to 0.5 s). In addition, the relative error defined as the normalised difference between two successive values of maximum impact pressure would be plotted. An optimum filtering interval would be defined as that with a relative error of 5% or lower. The same would then be carried out for the different DEM simulations with different parameters The filtering interval to be used for filtering the DEM and deducing the maximum impact pressure would be the optimal one while considering the two points above.

4) In our laboratory experiments finer grain size flow are faster than coarser ones (Cagnoli and Romano, 2012a). I therefore wonder whether the presence of an interstitial mud reduces the differential between the energy dissipation rates of flows with different grain size since you obtain virtually the same speed for them (lines 8-9, pag. 18). Unless the distance between position 2 and the release location is too small to see any difference.

The effect of the ratio of k1/k2 on the flow height and velocity was found to be limited because the variation of this ratio only concerns the interaction between the particles themselves. The interaction between flowing particles and the base of the channel is governed by a visco-elastic contact law where the value of epsilon_n was fixed based on previous studies of Albaba et al (2015). In that study, it was shown that the flow height is affected by the value of restitution coefficient while the flow velocity is governed by the value of the microscopic friction angle. All these details will be presented in detail in the new version.

5) Concerning the negative pressure values visible in Fig. 22 on pag. 23, are they artifacts due to pressure plate oscillations after collisions with the rock fragments?

The negative pressure values are artifacts of the sensor used in the experiments of Bungion et al 2012.

---

## Author Response (AR1)

Please refer to the previous authors' responses files (AC1 and AC2) for details concerning the response to all important points raised by the reviewers. The main changes in the article are highlighted in yellow.

[revised manuscript text omitted]